# Lymphocyte infiltration and thyrocyte destruction are driven by stromal and immune cell components in Hashimoto's thyroiditis

Qian-Yue Zhang [1,5], Xiao-Ping Ye [1,5✉], Zheng Zhou[1,4,5], Chen-Fang Zhu[2,5], Rui Li[1], Ya Fang[1], Rui-Jia Zhang [1], Lu Li[3], Wei Liu[3], Zheng Wang[1], Shi-Yang Song[1], Sang-Yu Lu[1], Shuang-Xia Zhao[1], Jian-Nan Lin [1✉] & Huai-Dong Song [1,3✉]

Hashimoto's thyroiditis (HT) is the most common autoimmune disease characterized by lymphocytic infiltration and thyrocyte destruction. Dissection of the interaction between the thyroidal stromal microenvironment and the infiltrating immune cells might lead to a better understanding of HT pathogenesis. Here we show, using single-cell RNA-sequencing, that three thyroidal stromal cell subsets, ACKR1+ endothelial cells and CCL21+ myofibroblasts and CCL21+ fibroblasts, contribute to the thyroidal tissue microenvironment in HT. These cell types occupy distinct histological locations within the thyroid gland. Our experiments suggest that they might facilitate lymphocyte trafficking from the blood to thyroid tissues, and T cell zone CCL21+ fibroblasts may also promote the formation of tertiary lymphoid organs characteristic to HT. Our study also demonstrates the presence of inflammatory macrophages and dendritic cells expressing high levels of IL-1β in the thyroid, which may contribute to thyrocyte destruction in HT patients. Our findings thus provide a deeper insight into the cellular interactions that might prompt the pathogenesis of HT.

[1] Department of Molecular Diagnostics & Endocrinology, The Core Laboratory in Medical Center of Clinical Research, State Key Laboratory of Medical Genomics, Shanghai Ninth People's Hospital Affiliated to Shanghai Jiao Tong University School of Medicine, Shanghai 200011, China. [2] Department of General Surgery, Shanghai Ninth People's Hospital, Shanghai Jiao Tong University School of Medicine, Discipline Construction Research Center of China Hospital Development Institute, Shanghai Jiao Tong University, Shanghai 200011, China. [3] Department of Endocrinology, Shanghai Ninth People's Hospital Affiliated to Shanghai Jiao Tong University School of Medicine, Shanghai 200011, China. [4] Present address: Department of geriatric endocrinology, the First Affiliated Hospital of Zhengzhou University, Zhengzhou 450000 Henan, China. [5] These authors contributed equally: Qian-Yue Zhang, Xiao-Ping Ye, Zheng Zhou, Chen-Fang Zhu. ✉email: yyfp007@126.com; linjiannan0111@163.com; huaidong_s1966@163.com

Autoimmune thyroid diseases (AITD), mainly Hashimoto's thyroiditis (HT) and Graves' disease (GD), are a consequence of loss of immune tolerance toward organ-specific self-antigens, including thyroglobulin (TG), thyroperoxidase (TPO), and the receptor for TSH[1]. HT is characterized by the infiltration of lymphocytes in thyroid tissues and the presence of autoantibodies against TPO and/or TG that are associated with the destruction of the thyroid follicles, which leads to hypothyroidism in a chronic progressive manner[1].

AITD is one of the most common autoimmune disorders, with a prevalence up to 5% and showing an increasing trend[2,3]. HT is related to higher risk of papillary thyroid carcinoma[4] and other autoimmune disorders such as type 1 diabetes, systemic lupus erythematosus, and rheumatoid arthritis (RA)[5–7]. The standard treatment for patients with hypothyroidism is thyroid hormone replacement therapy. However, a substantial proportion of patients treated with levothyroxine have persistent symptoms such as depression and impaired mental wellbeing, despite reaching the biochemical therapy targets[8], highlighting the need to explore alternative therapies.

Triggered by environmental and genetic factors, immune tolerance breaks down and leads to anti-thyroid autoimmune responses[9]. The infiltration of lymphocytes, especially highly organized lymphocytic infiltrates resembling secondary lymphoid organs (SLOs), termed tertiary lymphoid organs (TLOs) or ectopic lymphoid structures, often develops in the targeted tissues of patients with autoimmune diseases, including thyroid tissues in HT, synovium of joints in RA, and brain tissues in multiple sclerosis[10]. These tissues typically contain various hematopoietic cell types, high endothelial venules (HEVs), and follicular dendritic cells, and they are organized in lymph node-like structures[10]. Secondary lymphoid organogenesis involves recruitment of CD4+CD3−IL-7RA+ lymphoid tissue inducer (LTi) cells to sites of lymphoid organ development. The binding of lymphotoxin (LT) α1β2, expressed by LTi cells, to LTβR on stromal organizer cells induces expressions of *CCL19*, *CCL21*, *CXCL13*, and various adhesion molecules in organizer cells, which further promotes lymphocyte recruitment to SLO sites[11]. Notably, *CCL21*, *CXCL13*, *CXCL12*, and *CCL22* are increased in thyroid tissues in patients with HT[12]. Moreover, Furtado et al. found mice overexpressing the chemokine *CCL21* in the thyrocytes induced similar lymphocyte infiltration dependent on LTβR in thyroid tissues[13]. The mechanisms that regulate the formation, maintenance, and function of TLOs in inflamed tissues are largely shared with SLOs[14,15]. Thus, we presumed that the formation of TLOs in the thyroid tissues of HT patients might be similar to the formation of SLOs, which is mediated by the interaction of LTi cells with organizer cells in the stroma. However, the expression patterns of these chemokines in stromal organizer cells of the thyroid tissues in HT patients are not entirely clear.

It is well known that a series of cytokines and chemokines derived from infiltrated immune cells, such as TNF-α, IFN-γ, and IL-1β, are involved in the pathogenesis of HT. Thyroid follicular cells (TFCs) are induced by these cytokines to express surface molecules such as *Fas, MHC II, CD40,* and *ICAM*, and they further secret cytokines such as IL-1β, IL-6, IL-12, IL-13, and IL-18, perpetrating the destruction of thyrocytes and amplifying autoimmune responses in thyroid tissues of HT patients[16]. However, complete resolution of the cells is difficult, and bias is possible due to contamination of a purified cell population or growth in vitro.

Here we show the complex cell populations in thyroid tissues and peripheral blood mononuclear cells (PBMCs) from five HT patients using single-cell RNA-sequencing (scRNA-seq). We find specific stromal microenvironments that are critical for lymphocyte infiltration and organization in thyroid tissues of patients with HT. We also find thyroid-specific myeloid cell (MC) subsets related to the pathogenesis of HT. Our findings provide key insights into the mechanisms underlying the pathogenesis of HT.

## Results

**Single-cell profiling and unbiased clustering of cells in thyroid tissues of HT patients.** We isolated and sequenced all the cells from thyroid tissues and PBMCs derived from five HT patients (HT12, HT13, HT14, HT28, and HT29) (Fig. 1a). Among these HT patients, PBMCs of HT13 were not available, and scRNA-seq data of HT14 thyroid tissue contained too few cells to be used for further analysis due to technical mistakes. Patient information is listed in supplementary table 1. After exclusion of red blood cells, doublets, and cells not passing quality control (Supplementary Fig. 1B, C), 50,535 cells in thyroid tissues and 48,822 cells from PBMCs were analyzed.

We first analyzed the single-cell transcriptomes of all viable cells in the thyroid tissues. After correcting batch effects by "RunHarmony" function in the harmony package, cells from the four thyroid samples were clustered jointly. Unsupervised clustering analysis identified ten distinct cell clusters (Fig. 1b). All clusters included cells from multiple patients, suggesting that cells were grouped without patient specificity (Supplementary Fig. 1D).

To define the identity of each cell cluster, we generated cluster-specific marker genes by performing differential gene expression analysis using Monocle3 (Supplementary Data 1). The cell clusters were then manually classified into known cell-type populations based upon their differential expression of known and novel marker genes. Characterization of markers in these 10 clusters (Fig. 1c) identified them as four immune components and five stromal-follicular components, including T&NK cells (TCs), B cells (BCs), MCs, plasma cells (PCs), endothelial cells (ECs), ACTA2+ cells (ACs), fibroblasts (F), lymphatic endothelial cells (LEC), and TFCs as well as one group of proliferative cells (PrCs), which was then proved to be composed of mostly BCs and few TCs and plasmablasts (Fig. 1b). The groups could be distinguished using single marker genes, such as *CD3D* for TCs, *MS4A1* and *CD79A* for BCs, *CD138* (*SDC1*) and CD38 for PCs, *VWF* and *CD34* for ECs, *ACTA2* and *CNN1* for ACTA2+ stromal cells, CD14 and CD68 for MCs, MKI67 and TOP2A for PrCs, DCN and COL1A1 for fibroblasts, TG and TPO for TFCs, and PROX1 and SDPR for lymphatic ECs (Fig. 1c). Supplementary Data 1 lists the top maker genes for each of the ten clusters described in this study.

The fractions of the ten major compartments were estimated using scRNA-seq data in individual samples (Supplementary Fig. 1E). The infiltrated immune cells accounted for 65 to 82% of all the cells in thyroid tissues of HT patients, most of which were TCs, BCs, and PCs. HT12 had the lowest number of PCs and the most BCs, while HT28 has the highest number of PCs. These results are consistent with the clinical characteristics that HT12 has the lowest while HT28 the highest level of autoimmune antibodies (Supplementary Table 1).

**High-resolution analyses identified proinflammatory subgroups of stromal cells that may contribute to immune cell infiltration into thyroid tissues of HT patients.** To further address the transcriptional patterns of each cluster, cells in thyroid tissues were clustered into subgroups (Fig. 1d, Supplementary Data 1 and Data 2). TCs were classified into CD4+ TCs, CD8+ TCs, and NK cells; BCs were clustered into naive B cells (NaBs) and germinal center B cells (GCB) cells; Myeloid cells were divided into macrophages (Mac) and dendritic cells (DC); ACTA2+ cells (ACs) and fibroblasts each had two subgroups (AC1 and AC2; F1 and F2); ECs consisted of three subgroups

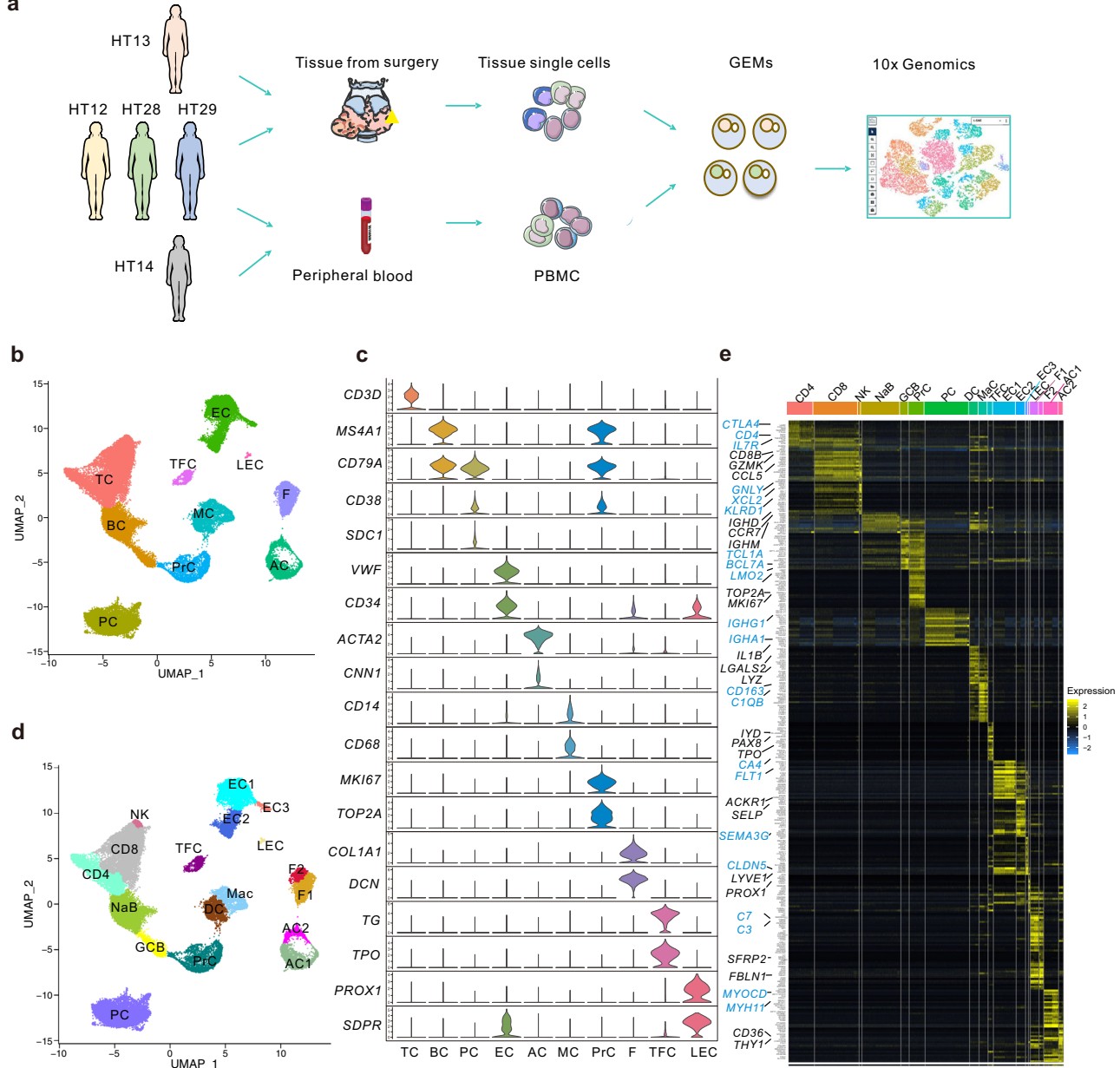

**Fig. 1 Single-cell RNA sequencing and clustering of cells from the thyroid of HT patients. a** Schematic showing sample workflow from operating room to sequencing. **b** UMAP of total cells from the thyroid tissues of HT patients after quality control, colored by cluster identity. **c** Violin plots showing the expression levels of representative marker genes across the clusters. The y axis shows the normalized read count. **d** UMAP of cells from the thyroid tissues of HT patients, colored by subgroup identity. **e** Single-cell expression heatmap displaying selected top marker genes of per cell cluster based on differential expression testing. TCs T cells, BCs B cells, PrCs proliferative cells, PCs plasma cells, MCs myeloid cells, TFCs thyroid follicular cells, ECs endothelial cells, LECs lymphocytic vessels endothelial cells, F fibroblasts, AC ACTA2⁺ cells, DC dendritic cells, Mac macrophages. NaB naive B cells, GCB germinal center B cells.

(EC1, EC2, and EC3) (Fig. 1d). Each subgroup showed distinct expression profiles, as depicted by a heatmap analysis of their marker genes (Fig. 1e).

Further analysis of subclusters and their signature genes revealed that subgroup AC2 of ACs expressed high levels of collagen-encoding genes such as *COL3A1*, cell adhesion molecule *CD36*, and *THY1*, a widely used marker for hematopoietic stem cells (Fig. 2a). However, subgroup AC1 specifically expressed *MYH11*, a smooth muscle cell marker that encodes a major contractile protein, myosin (Fig. 2a). Thus, we annotated AC2 as myofibroblast and AC1 as smooth muscle cells. Interestingly, both myofibroblasts (AC2) in the AC cell group and subgroup F1

in fibroblasts shared an expression profile of chemotactic factors, including chemokines for migration of MCs (*CCL2*)[17], chemokines for extensive leukocyte migration (*CXCL12*)[18], and constitutive lymphoid tissue-homing chemokines *CCL19* and *CCL21* (Fig. 2a, b). F1 also expressed high levels of IFN-γ-inducible TC trafficking chemokine *CXCL9*[19], BC survival-related gene *TNFSF13B* (*BAFF*)[20], and adhesive molecular *VCAM1*.

We further applied functional annotation on these stromal cell subgroups to reveal their different functions. Consistently, gene expressions of both F1 and AC2 subgroups were enriched in blood vessel development, the regulation of cell adhesion, cell migration or chemotaxis and cytokine-related responses or

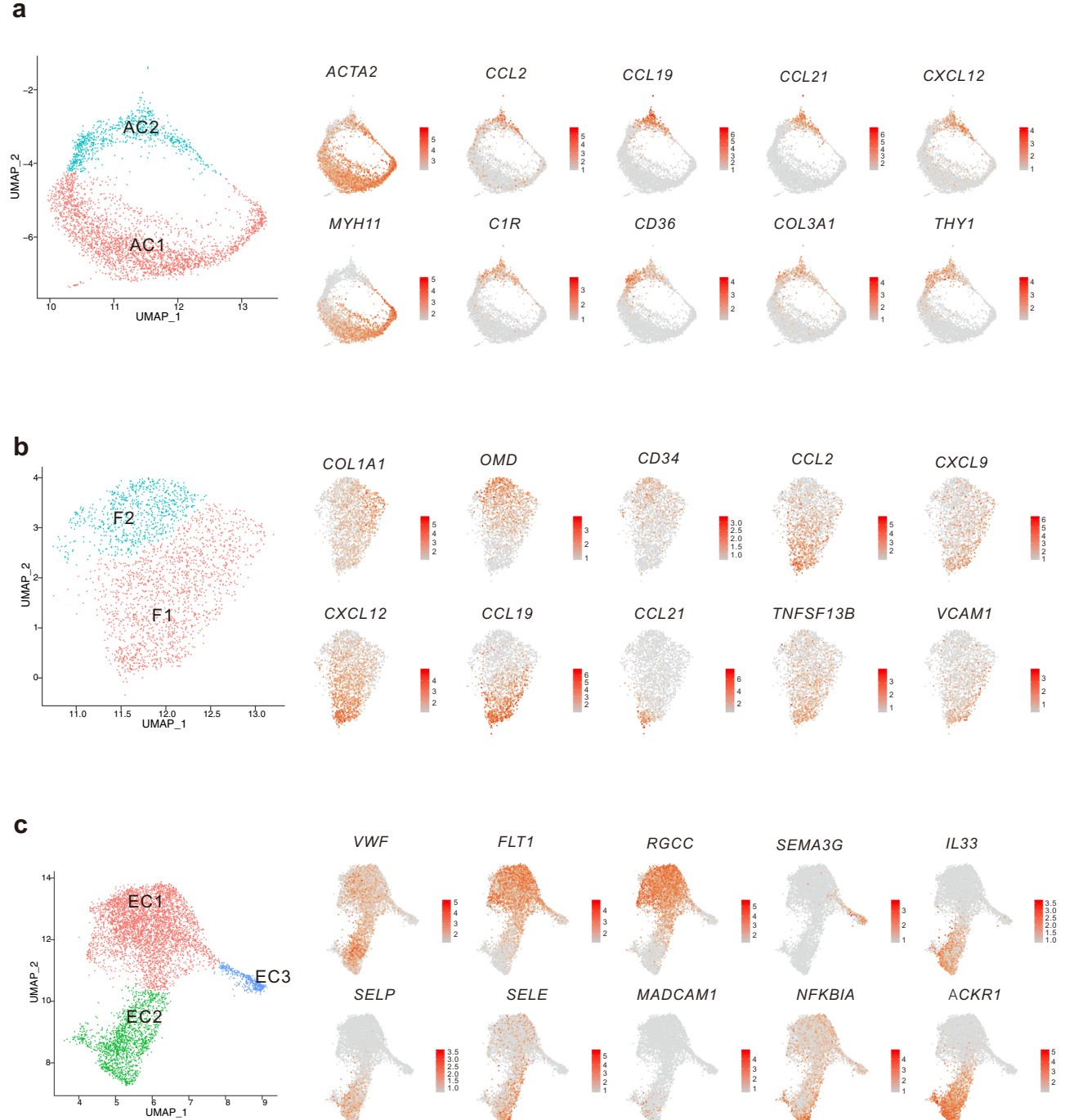

**Fig. 2 Inflammatory stromal cell subgroups and ACKR1⁺ HEV endothelial cells were identified in the thyroids of HT patients.** Marker gene expression of ACTA2⁺ cell subgroups (**a**), fibroblast subgroups (**b**), and endothelial cell subgroups (**c**) overlaid on the UMAP visualization.

pathways (Supplementary Fig. 2A, B). While AC1 was associated with muscle cell differentiation (Supplementary Fig. 2A). These results suggested myofibroblasts (AC2) and subgroup F1 fibroblasts may play critical roles in the infiltration of immune cells in HT by secreting chemokines and upregulating adhesion-related molecules.

ECs were further subclustered into three groups. EC1 expressed high levels of genes involved in blood vessel development such as *RGCC* and *FLT1* (Fig. 2c, Supplementary Fig. 2C). *RGCC* was found to be highly and specifically expressed in human choriocapillaris but not arterial and venous ECs[21], which suggests EC1 are capillary ECs. EC2 restrictively expressed *ACKR1* (Fig. 2c), a multi-chemokine

receptor, the expression of which can distinguish venular from non-venular ECs[22]. Notably, EC2 also showed high expressions of *SELP* and *IL33* (Fig. 2c), which are reported to be preferentially expressed in ECs of HEVs[23], as well as high expressions of *NFKBIA* and adhesion-related genes such as *SELE* and *MADCAM1*, which are usually upregulated in HEV ECs. HEVs are specialized post-capillary venules with lymphocyte trafficking functions located in lymph nodes and other SLO[24], suggesting EC2 contains a substantial group of HEV ECs. Consistently, enrichment analysis also revealed EC2 was related to positive regulation of cellular component movement, leukocyte activation, and adhesion (Supplementary Fig. 2C). EC3 specifically expressed *SEMA3G* and other genes related to

vessel development and cell motility (Fig. 2c, Supplementary Fig. 2C). *SEMA3G* was reported to display pronounced preferential arterial expression in all organs[25], suggesting EC3 are arterial ECs.

**CellPhoneDB analysis revealed a potential pathogenic cell interaction network between immune cells and stromal cells in thyroid tissues.** We further investigated cell–cell communication in thyroid tissues of patients with HT by CellPhoneDB based on the expression of ligands, receptors, and their interactions in different subtypes of cells[26,27] (Supplementary Data 3). Then, network analyses between stromal cells and immune cells, within stromal cells, and within immune cells were separately visualized using iGraph package in R (Supplementary Fig. 3A–C).

Interestingly, frequent cell interactions in HT thyroid tissues were found between stromal cells and immune cells, as depicted by the number of connecting lines between cell clusters (Supplementary Fig. 3A–C). Notably, the strongest interactions between stromal and immune subgroups involving immune cell trafficking were observed between the endothelial subgroup EC2, Fibroblast subgroup F1, myofibroblasts (AC2), and the immune cells (Fig. 3a). In detail, EC2 expressed high levels of *ACKR1*, which is a shared receptor for many chemokines such as *CCL2* and *CXCL8* expressed on dendritic cells and macrophages, as well as *CCL5* expressed in all subgroups of immune cells (Fig. 3a, b). The interactions of myofibroblasts (AC2) and fibroblast subgroup F1 with immune cells were mediated by *CCL19*, *CCL21*, and their common receptor *CCR7* (Fig. 3a, b). *CCR7*, expressed on TCs, dendritic cells, and naive and proliferative BCs, is one of the most important chemokine receptors for adaptive immune cell migration. Its ligands with *CCL19* and *CCL21*, which were abundantly expressed on stromal cells such as myofibroblasts (AC2) and fibroblasts (F1) in thyroid tissues of HT patients, control homing of TCs and dendritic cells to lymph node areas[28]. *CCL21* was also expressed by LECs (Fig. 3a), which has been reported to cause dendritic cell chemotaxis toward lymphatic vessels[29]. Our findings suggest, by production of a series of chemokines, that stromal cells such as EC2, AC2, and F1, rather than thyrocytes, may play key roles in promoting the infiltration of immune cells in thyroid tissues of HT patients.

In addition, we also found several interactions of adhesive and pro-adhesive molecules between vascular ECs and immune cells in thyroid tissues of HT patients (Fig. 3a). The adhesion of dendritic and NK cells with ECs might be mediated through the interaction of *ICAM1* with *AREG*, but macrophage, DC, NK, and most BCs adhered to ECs by the interaction of *PECAM1* with *CD38*. Meanwhile, all the immune cells except PCs also adhered to ECs through the interaction of *SELL* expressed on the immune cells, with *CD34* and *PODXL* expressed on ECs. Notably, the adhesion of immune cells to EC2 ECs might be mediated by the especially strong interaction of *SELE* and *SPP1* in EC2 cells with *CD44* expressed on all the immune cells. The interaction of *SELP* in EC2 cells with *CD24* expressed on BCs and macrophages might merely mediate the adhesion of BCs and macrophages to EC2 cells in HT thyroid tissues. Previous studies suggested that expression of adhesion molecules such as *SELP* and *SELE* on the endothelium was a critical early step for leukocyte migration from venules to tissues in AITD[30,31]. Our findings suggested several adhesion molecules are upregulated in ECs, particularly the EC2 subgroup, in thyroid tissues of HT patients and could promote the adhesion and recruitment of immune cells to the thyroid tissue.

**Immunostaining revealed specific stromal microstructures correlated with lymphocytic aggregates or tertiary lymphoid neogenesis.** We found that atypical chemokine receptor-1 (*ACKR1*) was restrictively and highly expressed in EC2 cells of thyroid tissues

from HT patients (Fig. 2c, Supplementary Fig. 3D). Interestingly, previous studies have reported that *ACKR1*, expressed in the ECs of HEV, promoted lymphocytic infiltration from the vessels by binding to several proinflammatory CC and CXC chemokines[32,33]. By multicolor immunostaining of ACKR1, vascular EC marker VWF, and leukocyte marker CD45, we found small vessels with a compressed morphology and a luminal diameter of 10–100 μm, were remarkably increased in the region with abundant infiltrated lymphocytes in thyroid tissues of HT patients (Fig. 4a). The ECs of these veins are ACKR1[+]. Moreover, in areas with less immune cell infiltrates and thyrocyte destruction of HT thyroid tissues, CD45[+] leukocytes were suggested to be undergoing trans-endothelial migration in ACKR1[+] venules (Fig. 4b). However, no ACKR1-staining in ECs was observed in capillaries with a diameter of 7–9 μm (Fig. 4a, b; blue arrow). Further analysis indicated that ACKR1[+] ECs were also absent in arteries with thick layers of smooth muscle cells by multicolor immunostaining with ACTA2, a common marker for smooth muscle cells and myofibroblasts (Fig. 4c; blue arrow). By double-color immunostaining of MECA-79 (a specific antibody to mark HEVs) and ACKR1, we found that all the ECs of MECA-79[+] HEVs expressed ACKR1, but there were still some ACKR1[+] venules negative for MECA-79 (Fig. 4d; blue arrow).

In our scRNA-seq data of thyroid tissues, we found the AC subgroup AC2 specifically expressed high levels of chemokines such as *CCL2*, *CXCL12*, *CCL19*, and *CCL21* (Fig. 2a). Thus, we further investigated the distribution of AC2 cells in thyroid tissues from HT patients. We noted the expression of *CD36* in AC2 cells was higher than that in AC1 cells in thyroid tissues from HT patients (Fig. 2a); thus, CD36 could be used as a marker to distinguish AC2 (myofibroblasts) from AC1 (smooth muscle cells). Interestingly, myofibroblasts (double-positive for CD36 and ACTA2) were merely distributed in part of the ACKR1[+] venules, which were surrounded by a thin layer of ACTA2 and CD36 double-positive myofibroblasts (Fig. 4e). However, the cells double-positive for CD36 and ACTA2 were not distributed in the arteries, which were surrounded by a thick layer of ACTA2[+] smooth muscle cells in thyroid tissues of HT patients (Fig. 4e).

Aside from myofibroblasts, *CCL21* was also expressed on fibroblast subgroup F1 (Fig. 2b) and lymphatic vessels ECs (Supplementary Fig. 3D). We further clarified the location of *CCL21[+]* stromal cells in thyroid tissue of HT patients using multicolor staining with the *CCL21* RNA probe and antibodies of protein markers for lymphocytes (CD45), BCs (CD20), TC (CD3E), blood vascular ECs (VWF), myofibroblasts or fibroblasts (COL1A1), and myofibroblasts or smooth muscle cells (ACTA2). *CCL21[+]* cells showed three distribution patterns in thyroid tissues of HT patients. Some *CCL21[+]* cells were observed in the adventitia of MECA-79[+] HEVs and recognized as *CCL21[+]* myofibroblasts (double-positive for *CCL21* and ACTA2) (Fig. 5a, b), which is consistent with the result in Fig. 4e. Some *CCL21[+]* cells were observed in lumen structures different from blood vessels and were recognized as lymphatic vessels ECs (Fig. 5c). Interestingly, we found most of the *CCL21[+]* cells were distributed in TC zones of TLOs in thyroid tissues of HT patients (Fig. 5d), in which *CCL21* was co-expressed with COL1A1 (Fig. 5e) but not with ACTA2 (Fig. 4f); thus, the *CCL21[+]* cells were defined as *CCL21[+]* fibroblasts. In areas without germinal centers, *CCL21[+]* fibroblasts were scattered around HEVs and were surrounded by infiltrated lymphocytes (Fig. 5g). These findings indicated that *CCL21[+]* fibroblasts may play key roles in recruiting lymphocytes into thyroid tissues of HT patients.

ACKR1 is reported to regulate the activity of chemokines including CCL2 and CCL5, by translocating them across the endothelial barrier after its binding to them[32,33]. Thus, we speculated that the aberrant expression of *ACKR1* in EC2 may contribute to lymphocyte infiltration in HT thyroid tissues. This was supported by

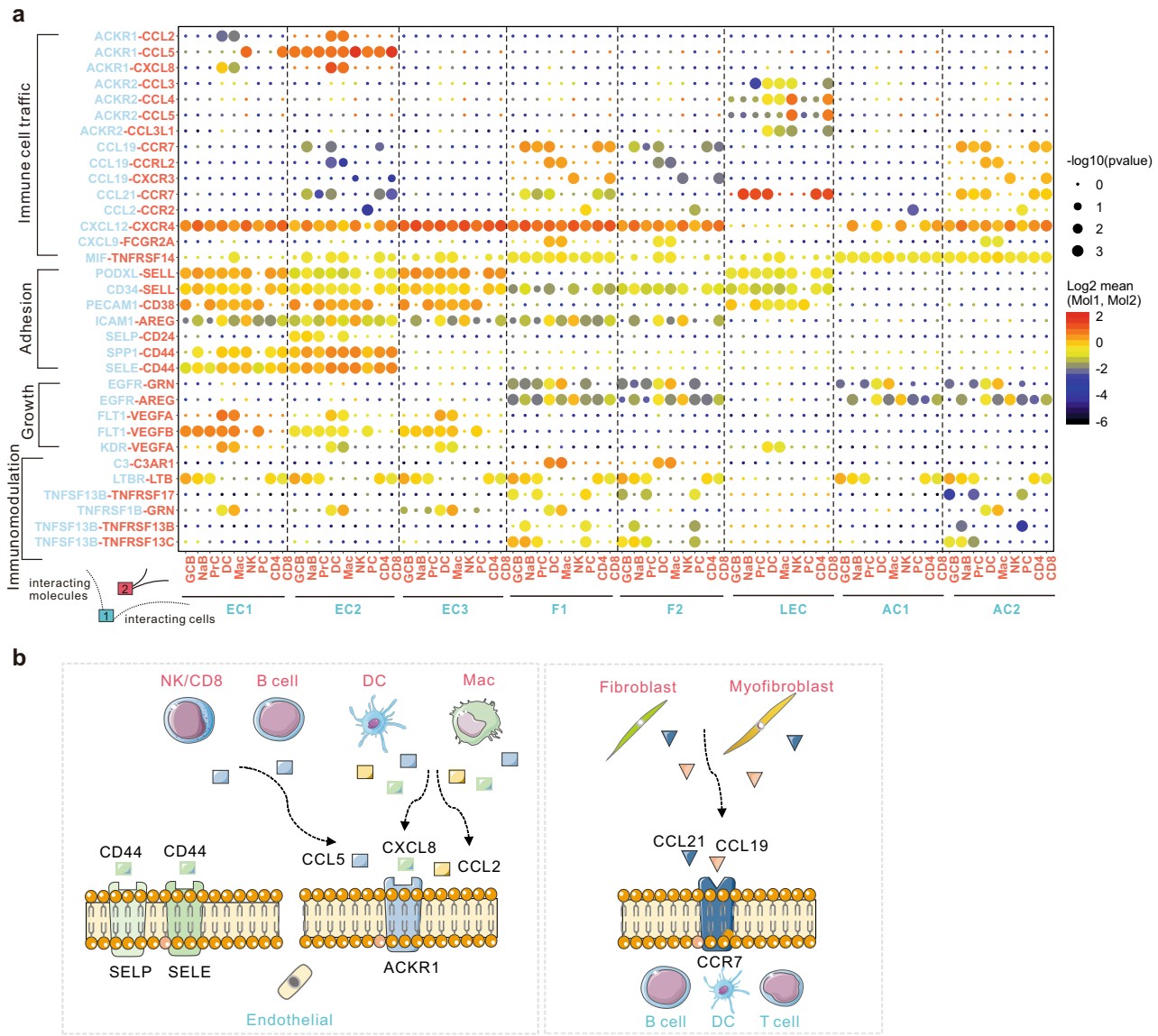

**Fig. 3 Extensive cell interactions that may facilitate immune cell trafficking exist between stromal cells and immune cells. a** Overview of selected ligand–receptor interactions between the thyroidal stromal cell subsets and immune cell clusters in thyroid tissues of HT patients. Circle size denotes corresponding $p$ value of one-sided permutation test with 10,000 permutations. The means of the average expression level of interacting molecule 1 in cluster 1 and interacting molecule 2 in cluster 2 are indicated by color. **b** Diagram of the main receptors and ligands expressed on the thyroidal stromal cells and immune cells that are involved in adhesion and cellular recruitment.

in vitro lymphocyte transmigration through *ACKR1*-overexpressing human umbilical vein endothelial cell (HUVEC) monolayers in response to chemokines. As expected, the overexpression of *ACKR1* in HUVECs significantly enhanced the transmigrating ability of PBMCs through HUVEC monolayer cells in the presence of ligands for ACKR1, such as CCL2 and CCL5, which are highly expressed by myofibroblasts and the F1 fibroblasts subset, CD8+ TCs, and NK cells, respectively (Supplementary Fig. 3D; Fig. 4f). Meanwhile, the in vitro transwell study also found CCL21 in the media of cultured C2C12 myoblasts transfected with mouse *ccl21* significantly promoted the migration of T8.1 TCs, a murine hybridoma TC line expressing CCL21 receptor CCR7 (Fig. 4g). These findings suggest that chemokines expressing myofibroblasts combined with ACKR1+ ECs in HEVs facilitated the infiltration of blood lymphocytes in venules into the thyroid tissues of HT patients.

To clarify whether these specific stromal cell subsets exist in non-HT controls, we compared the numbers of ACKR1+ vessels and *CCL21*+ fibroblasts or myofibroblasts in thyroid tissues

between HT patients and non-HT controls. Through immunostaining of FSP1 (Fibroblast-specific protein 1) and ACTA2 combined with in situ hybridization of *CCL21*, FSP1+*CCL21*+ fibroblasts or myofibroblasts are easily observed in germinal centers (Fig. 6a), area of lymphocyte aggregates (Fig. 6c) or adventitia of vessels (Fig. 6c, white arrows), as well as the area of intact thyroid follicular structures with few lymphocytic infiltration (Fig. 6b) in the thyroid tissues of HT patients. However, in non-HT controls, fibroblasts are rarely and *CCL21*+ cells are barely not observed (Fig. 6d). By immunostaining of MECA-79, VWF and ACKR1, we found MECA-79+ HEVs are not observed in thyroid tissues from non-HT controls (Fig. 6e, f). Moreover, the number of ACKR1+ vessels and the rate of ACKR1+ vessels are significantly decreased in the thyroid tissues of non-HT controls ((Fig. 6g, h) when compared with the patients with HT.

To further investigate the association of cell subsets identified in scRNA-seq and disease, differential gene expression analyses

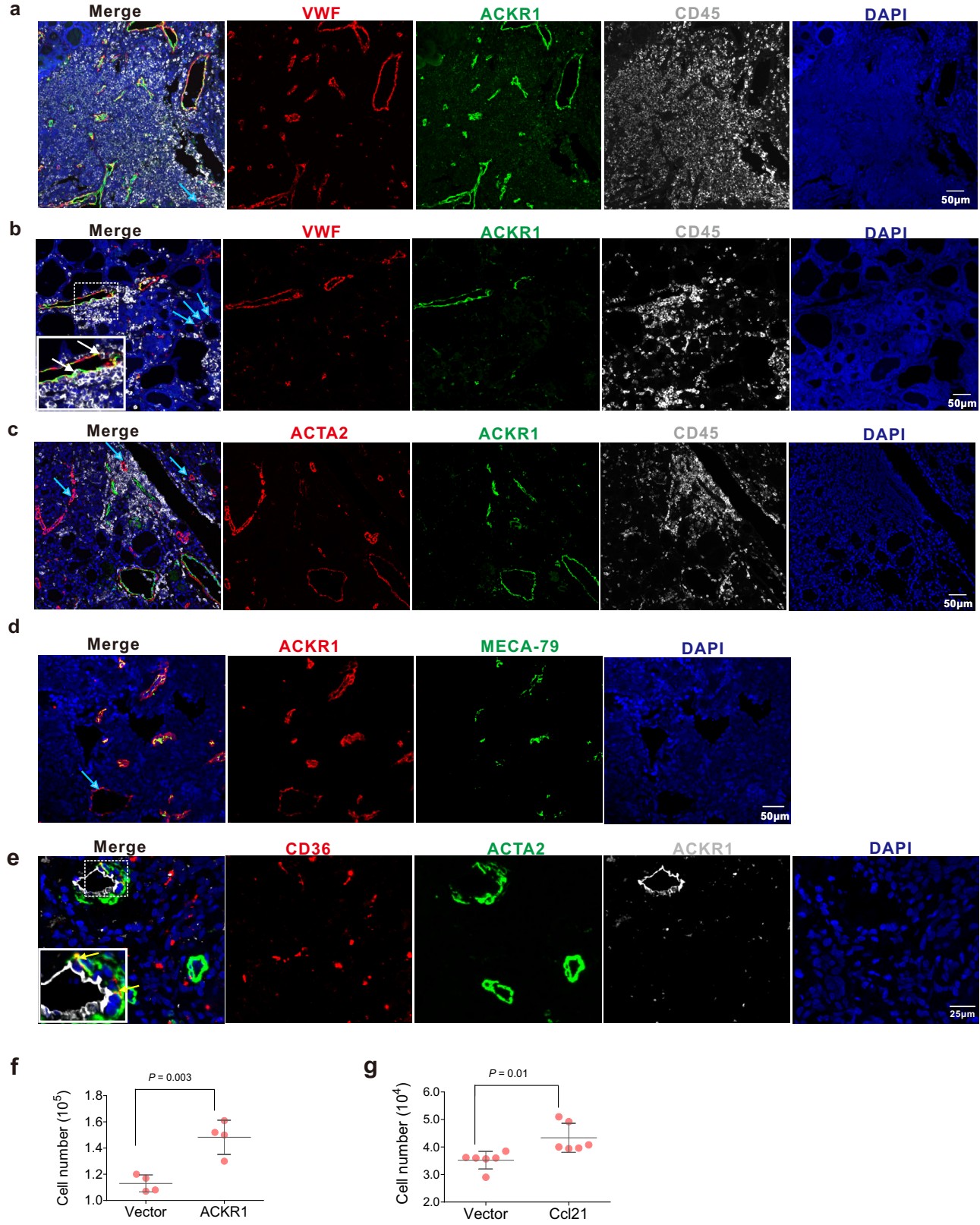

were performed by bulk RNA-seq on thyroid para-tumor tissues of 16 HT patients and 50 non-HT controls who underwent surgery due to thyroid tumor (Supplementary Data 4). As expected, the upregulated genes in HT thyroid tissues showed enriched GO process of lymphocyte activation, adaptive immune response, immune effector process, leukocyte migration, as well as cytokine production (Supplementary Fig. 4A). Interestingly, in addition to immune cell subgroups, the expression scores of the top markers for stromal cell subgroups EC2 and F1 identified by Monocle in the thyroid tissues of HT patients were significantly higher than those of non-HT controls, suggesting HEV EC2 ECs and F1 fibroblasts are increased in thyroid tissues of HT patients

**Fig. 4 Polychromatic immunofluorescence staining showing the distribution of myofibroblasts, ACKR1⁺ venules, and HEVs in thyroid tissues of HT patients. a** Immunostaining for ACKR1, pan-lymphocyte marker CD45, and endothelial cell marker VWF showing the distribution of ACKR1⁺ cells in the thyroids of HT patients. Large numbers of ACKR1⁺ vessels were observed in areas with numerous immune cell infiltrations. ACKR1 was not expressed by capillaries that had a diameter <10 µm (blue arrows) in the thyroid tissues of HT patients. **b** Immunostaining for ACKR1, CD45, and VWF suggesting lymphocyte migration through the vascular wall (white arrows). ACKR1 was not expressed by capillaries (blue arrows). **c** Immunostaining for ACKR1, pan-lymphocyte marker CD45, and smooth muscle cell marker ACTA2 shows ACKR1 is expressed in venules with compressed lumen structure but not arteries with thick smooth muscle layers (blue arrows). **d** Immunostaining for ACKR1 and MECA-79 (a specific antibody to mark the HEVs) shows most ACKR1⁺ venules are MECA-79⁺ HEVs. Blue arrow indicates MECA-79 negative venule, which also expressed the ACKR1. **e** Immunostaining for CD36, ACTA2, and VWF showing the existence of myofibroblasts (CD36 and ACTA2 colocalization) in the vascular structures of thyroid tissues from HT patients. Myofibroblasts were observed merely in ACKR1⁺ venules with thick endothelial and compressed lumen structure (yellow arrows) but not arteries with thick smooth muscle layers. **f** The HUEVC overexpressed ACKR1 cultured in the upper chamber promoted trans-endothelial migration of PBMCs under CCL2 and CCL5 chemotaxis (10 ng/ml) in lower chamber. Migrated PBMCs in the lower chamber were counted 20 h after seeding. two-sided Student's $t$ test, data are representative of three independent experiments. Each dot represents the number of cells in one well of the plate. The error bars indicate SD. **g** Transwell migration assay shows myofibrocytes transfected with mouse *ccl21* significantly promoted the migration of T8.1 T cells at 20 h after cell seeding. two-sided Student's $t$ test, data are representative of three independent experiments. Each dot represents the number of cells in one well of the plate. The error bars indicate SD.

(Supplementary Fig. 4B). Consistently, further investigation of specific markers for EC2 and F1 showed that the expression levels of *ACKR1*, *CCL19*, *CCL21*, *CXCL9*, *SELP*, *IL33*, and *MADCAM1* were significantly increased in thyroid tissues of HT patients (Supplementary Fig. 4C, Supplementary Data 4).

**Comparison of immune cells between thyroid tissues and PBMCs of HT patients revealed distinct proportions and phenotypes.** To investigate whether there were functional differences between immune cells present in the thyroid tissues and peripheral blood of HT patients, we further compared single immune cells between the thyroid tissue and PBMCs of HT patients. After mapping and quality control, the 10× genomics data of thyroidal immune cells were merged with PBMCs for further analysis. After correcting the batch effects using sample identity as covariates in the merged 10× genomics data of thyroidal immune cells and PMBCs, three major cell clusters were identified by clustering analysis including T/NK cells, MCs, and B/PC, as indicated by classic markers (Fig. 7a, Supplementary Fig. 5A). All clusters included cells from four patients, suggesting that cells were grouped without patient specificity (Supplementary Fig. 5B). Proportional calculation and separated view of cells originating from the thyroid or PBMCs showed obvious cell compositional differences of the three clusters (Fig. 7b, Supplementary Fig. 5B). There were more TCs and MCs in PBMCs than in thyroid tissues, while remarkably more B/PCs were found in thyroid tissues than in PBMCs. Moreover, significant differences were found in the distribution of monocyte/DC between PBMCs and thyroid tissues (Supplementary Fig. 5B).

The three major clusters were further clustered into subgroups, and cell types were defined based on specific marker genes. TCs were divided into four subgroups, including one group of proliferative TCs detected exclusively in the thyroid tissues of patients with HT. These proliferative TCs were clustered closely to proliferative BCs because of their high expression of proliferation-related genes (Fig. 7c, Supplementary Data 5). In addition, we found BCs were the major immune cell population in the thyroid tissues, and the ratio of CD8⁺ TCs to CD4⁺ TCs increased slightly in thyroid tissues compared to that in PBMCs, but NK cells decreased significantly (Fig. 7d). While CD4⁺ TCs in thyroid tissues of HT patients could be classified into Th1, Th2, or Th17 subtypes, as previously reported[34], CD4⁺ TCs did not segregate into distinct clusters in our study, and the expression of TC subset marker genes, such as *IFNG* and *TBX21* for Th1, *IL4* and *GATA3* for Th2, and *IL17RA* and *RORC* for Th17, were low and sporadic (Supplementary Fig. 5C). This may be due to the dropout events commonly seen in scRNA-sequencing. However, plots of classical TC subtype markers could identify naïve TCs (*CCR7*), Tregs

(*FOXP3* and *IL2RA*), and CD8⁺ cytotoxic TCs (*GZMB*), and *IFNG* expression was identified mainly in CD8⁺ TCs (Supplementary Fig. 5C). Notably, striking differences were found in the distribution of CD4⁺ and CD8⁺ TCs from PBMCs and thyroid tissues of HT patients (Fig. 7c). Moreover, *XCL1* and *XCL2* were predominantly expressed on CD8⁺ TCs from thyroid tissues compared with that from PBMCs of HT patients (Supplementary Fig. 5C). The X-C motif chemokine ligand (XCL) was reported to be expressed by memory CD8⁺ TCs and NK cells and facilitate interaction of cDC1 with these effector cells by binding with its receptor XCR1[35]. These results suggest altered functions were gained by TCs after infiltrating into the thyroid tissues of HT patients.

**Numerous tissue germinal center B cells and plasma cells indicate the thyroid as the major source of autoimmune antibodies.** The B/PC cluster was classified into eight subgroups (B1-B8) (Fig. 7e) according to their marker genes (Fig. 7g) and differential expression profiles (Supplementary Data 6). B1 expressed genes encoding *IgD* and *IgM* (*IGHD* and *IGHM*) and was defined as NaBs (Fig. 7g). B2 expressed moderate levels of *CD86* and *CD40*, indicating a more activated phenotype compared with NaBs. Notably, B2 showed specific expression of the classical DC marker *CD11C* (*ITGAX*), and it was defined as a CD11c⁺ BC (Fig. 7g), which has been reported to accumulate in aged females and autoimmune-prone mice and play a role in the development of autoimmunity[36]. Germinal centers (GCs) are usually formed in the thyroid tissues of HT patients[37]. Consistent with this, B3 was recognized as centrocytes in the light zone (LZ) because of the expression of *LMO2* (Fig. 7g), a GCB-specific gene[38]. B4 and B5 were identified as centroblast1 and centroblast2 in the dark zone because of the expression of proliferative markers *MKI67* and *TOP2A* (Fig. 7g). Compared with centroblast1 (B4), centroblast2 (B5) did not express *BCL6* and *SERPINA9*, and it presented merely in thyroid tissues of HT patients. *BCL6* is an anti-apoptosis gene critical for the survival of GCBs, and *SERPINA9* encodes protease inhibitors that are restrictively expressed in the germinal center of SLO[39], suggesting the apoptotic tendency of centroblast2. Both B6 and B7 expressed high levels of PC markers *PRDM1*(*BLIMP-1*), *SDC1*(*CD138*), and *XBP1*, which are typically upregulated in antibody-producing cells. Since B6 also expressed MKI67 and TOP2A, B6 and B7 were thus defined as plasmablasts and PCs, respectively (Fig. 7g). Both *BLIMP-1* and *CD138* expression levels are upregulated during PC maturation and are required for long-term survival of PCs[40–42]. Interestingly, we identified a novel group of antibody-producing cells (B8), which also expressed high levels of *XBP1* but low levels of plasma survival-related genes *BLIMP-1* and *CD138* (Fig. 7g). Thus, we

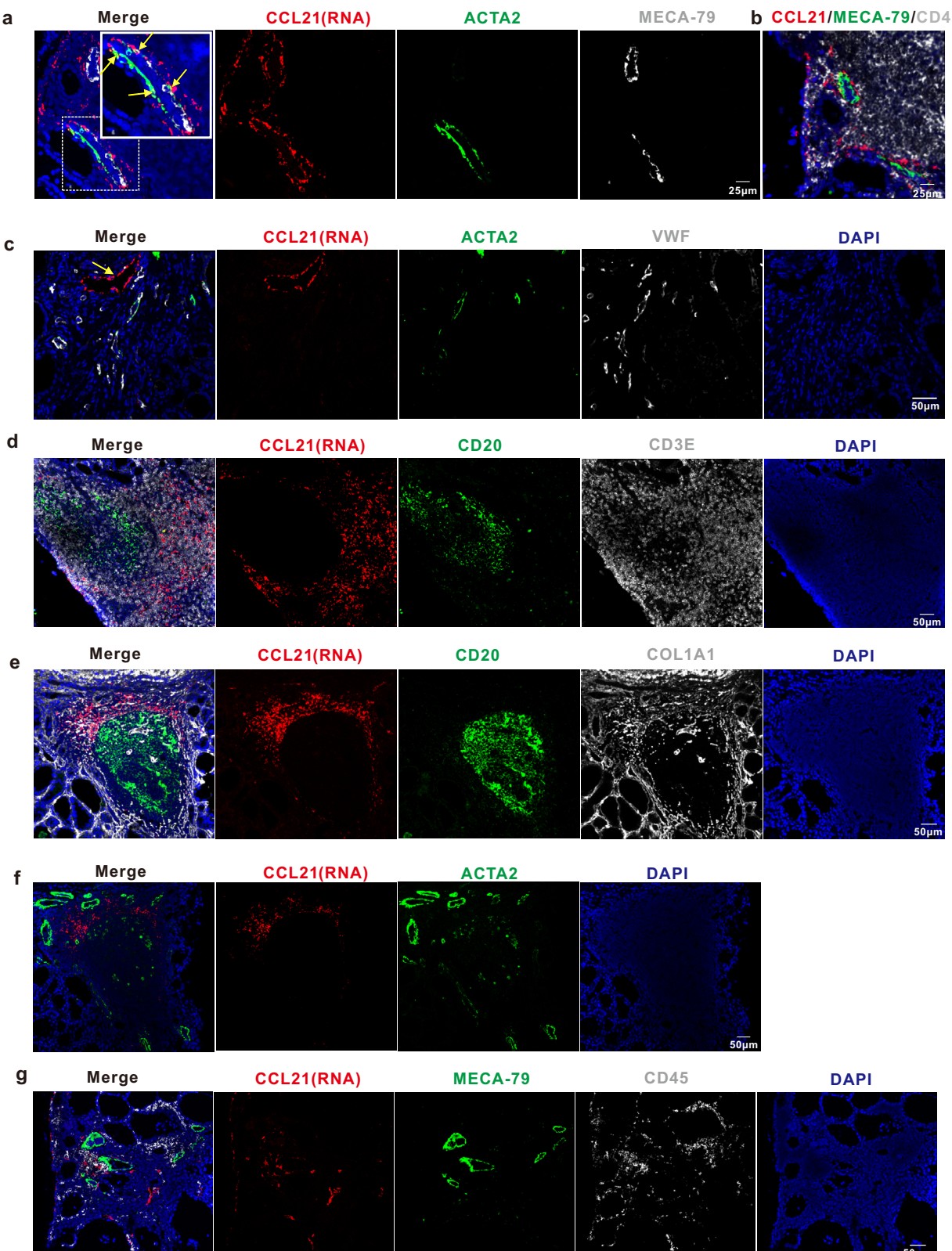

**Fig. 5 Distribution of CCL21⁺ stromal cells in thyroid tissues of HT patients suggests the critical role of CCL21⁺ fibroblasts in tertiary lymphoid organ organization. a** RNAscope in situ hybridization combined with polychromatic immunofluorescence staining showing the distribution of ACTA2⁺CCL21⁺ myofibroblasts in the adventitia of HEVs. The colocalization of CCL21 and ACTA2 (yellow arrows) indicate CCL21⁺ myofibroblasts in the adventitia of MECA-79⁺ HEVs. **b** Adjacent slice of **a** shows MECA-79⁺ HEVs are surrounded by many lymphocytes. **c** The distribution of CCL21⁺ lymphatic vessels endothelial cells. CCL21 signal shows a lumen structure negative of VWF and ACTA2. **d**–**f** Large numbers of CCL21⁺ fibroblasts are distributed in the T cell zone of the TLOs. CD20 indicates B cells and CD3E indicates T cells. Section in **f** is the adjacent slice of (**e**). CCL21⁺ cells are positive for COL1A1 but negative for ACTA2. **g** In areas with less lymphocytic infiltration, all CCL21⁺ fibroblasts are around lymphocytes.

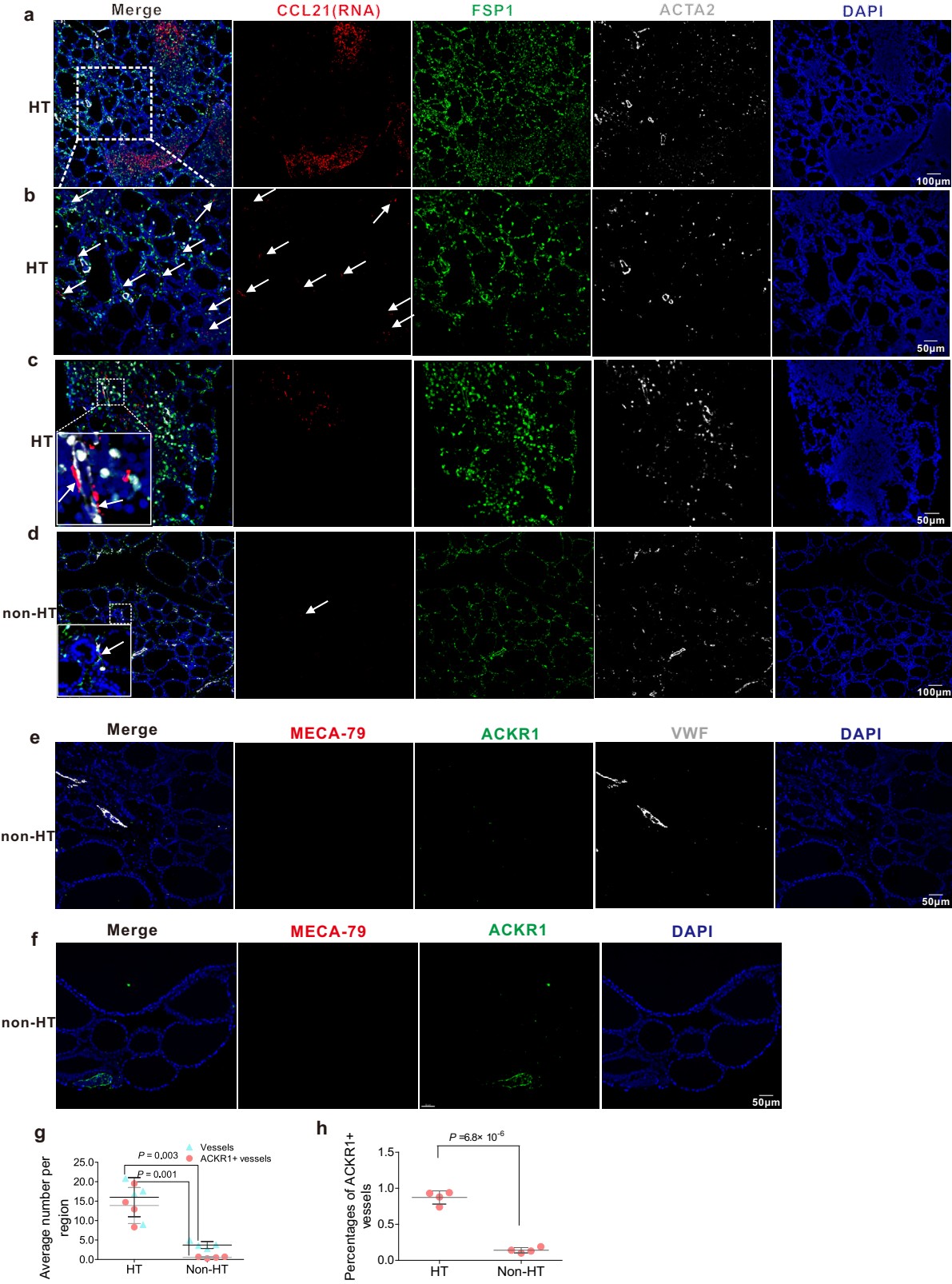

speculate B8 might be short-lived or senescent PCs, while B7 were long-lived PCs[43]. By comparing the proportion of these subgroups between thyroidal and PBMCs BCs, we found PCs B7 and plasmablasts B6 were markedly increased, and NaBs were dramatically decreased, in thyroid tissues of HT patients compared with PBMCs (Fig. 7f). Taken together, the presence of GCBs, plasmablasts, and numerous long-lived PCs in thyroid tissues of

HT patients suggested that autoantibodies against thyroidal antigens were produced by thyroid-derived PCs.

**Inflammatory DCs and macrophages are present in thyroid tissues but not in PBMCs of patients with HT.** By subcluster analysis, MCs from the thyroid tissues and PBMCs of HT patients

**Fig. 6 ACKR1$^+$ endothelial cells and *CCL21*$^+$ fibroblasts or myofibroblasts are few in thyroid tissues of non-HT controls compared with HT patients.** **a–d** *CCL21*$^+$ fibroblasts or myofibroblasts are significantly increased in thyroid tissues of HT patients compared with non-HT controls. In situ hybridization combined with polychromatic immunofluorescence staining shows FSP1$^+$*CCL21*$^+$ fibroblasts or myofibroblasts are easily observed in germinal centers (**a**), area of lymphocyte aggregates (**c**) or adventitia of vessels (**c**, white arrows), as well as the area of intact thyroid follicular structures with few lymphocytic infiltration (**b**) of the thyroid tissues of HT patients. While in thyroid tissues of non-HT controls, fibroblasts are rarely and *CCL21*$^+$ cells are barely not observed (d). FSP1 is the name for fibroblast-specific protein 1. Panel **b** is magnified image of the area of rectangular box in (**a**). **e-f** HEVs are not observed and ACKR1$^+$ vessels are significantly decreased in thyroid tissues of non-HT controls. Total number of vessels and ACKR1$^+$ vessels (**g**), and the rate of ACKR1$^+$ vessels (**h**) in thyroid tissues of non-HT controls are significantly less than that of in HT patients. 4 vs 4, 3–5 regions in each of the 3 different slices for each sample are stained and quantified. Two-sided Student's *t* test. The error bars indicate SD.

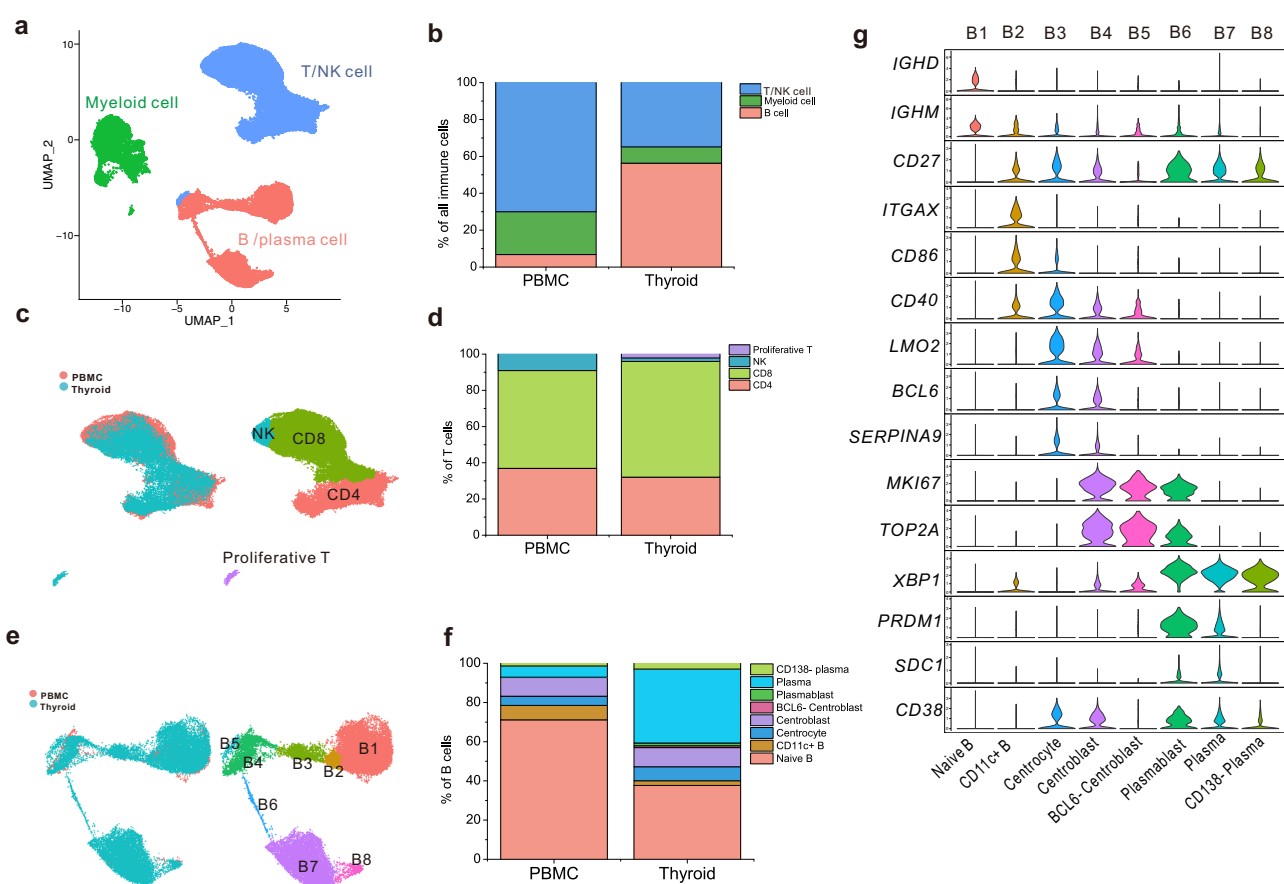

**Fig. 7 Clustering and cell proportion analysis of merged immune cells from the thyroid and PBMCs of HT patients.** **a** UMAP of immune cells from the thyroid and PBMC of HT patients, colored by cluster identity. **b** Total immune cells subgroup proportions in PBMCs and thyroid tissues of HT patients. **c** UMAP of merged T cells, colored by source of PBMCs or thyroids (left) and cluster identity (right). **d** Total T cell subgroup proportions in PBMCs and thyroid tissues of HT patients. **e** UMAP of merged B/plasma cells, colored by source of PBMCs or thyroids (left) and cluster identity (right). **f** Total B/plasma cell subgroup proportions in PBMCs and thyroid tissues of HT patients. **g** Violin plots showing the expression levels of representative marker genes across the clusters of B/plasma cells. The *y* axis shows the normalized read count.

were divided into seven subgroups, named M1-M7, including two groups of monocytes (M1 and M3), two groups of macrophages (M4 and M7), and three groups of DCs (M2, M5, and M6) (Fig. 8a, Supplementary Data 7). M1 and M3, which were dominant in the PBMCs of HT patients, abundantly expressed *CD14* or *FCGR3A* (*CD16*) and were considered as classical CD14$^+$ monocytes and non-classical CD16$^+$ monocytes, respectively (Fig. 8a–c). On the other hand, M2 and M4 were found to be dominantly present in thyroid tissues of HT patients (Fig. 8b), with high expressions of plasmacytoid dendritic cell (pDC) marker genes *CLEC4C* and *IL3RA*, and macrophage marker genes *CD163*, *C1QA*, and *C1QB*, respectively (Fig. 8c). Thus, M2 was defined as pDC and M4 as macrophages. Interestingly, M4 also expressed *CD14* and *CD16*, which are indicative of an intermediate monocyte phenotype[44].

Notably, M6 and M7 specifically existed in thyroid tissues rather than in PBMCs of HT patients (Fig. 8a). M6 expressed *CD1C* and *FCER1A*, the marker genes of conventional dendritic cell 2 (cDC2), and was considered as cDC2 (Fig. 8c), while M7 expressed *CD163*, *CD68*, and *CD14*, the marker genes of macrophages, and was defined as macrophages (Fig. 8c). Interestingly, the expression of inflammatory cytokine genes such as TNF-α-induced gene *G0S2*[45], *IL-1B* (encoding IL-1β), and TNF-α-induced apoptosis-associated gene *IER3* were higher in M6 than in M5, which also expressed the marker genes of cDC2, *CD1C*, and *FCER1A*, but they appeared in both thyroid tissues and PBMCs of HT patients (Fig. 8a, c). M7 expressed lower levels of macrophage marker genes but higher levels of inflammatory cytokines and chemokines such as *IL1B*, *CXCL2*, *CCL20*, and *IER3*[46] compared to the M4 subset. Thus, we

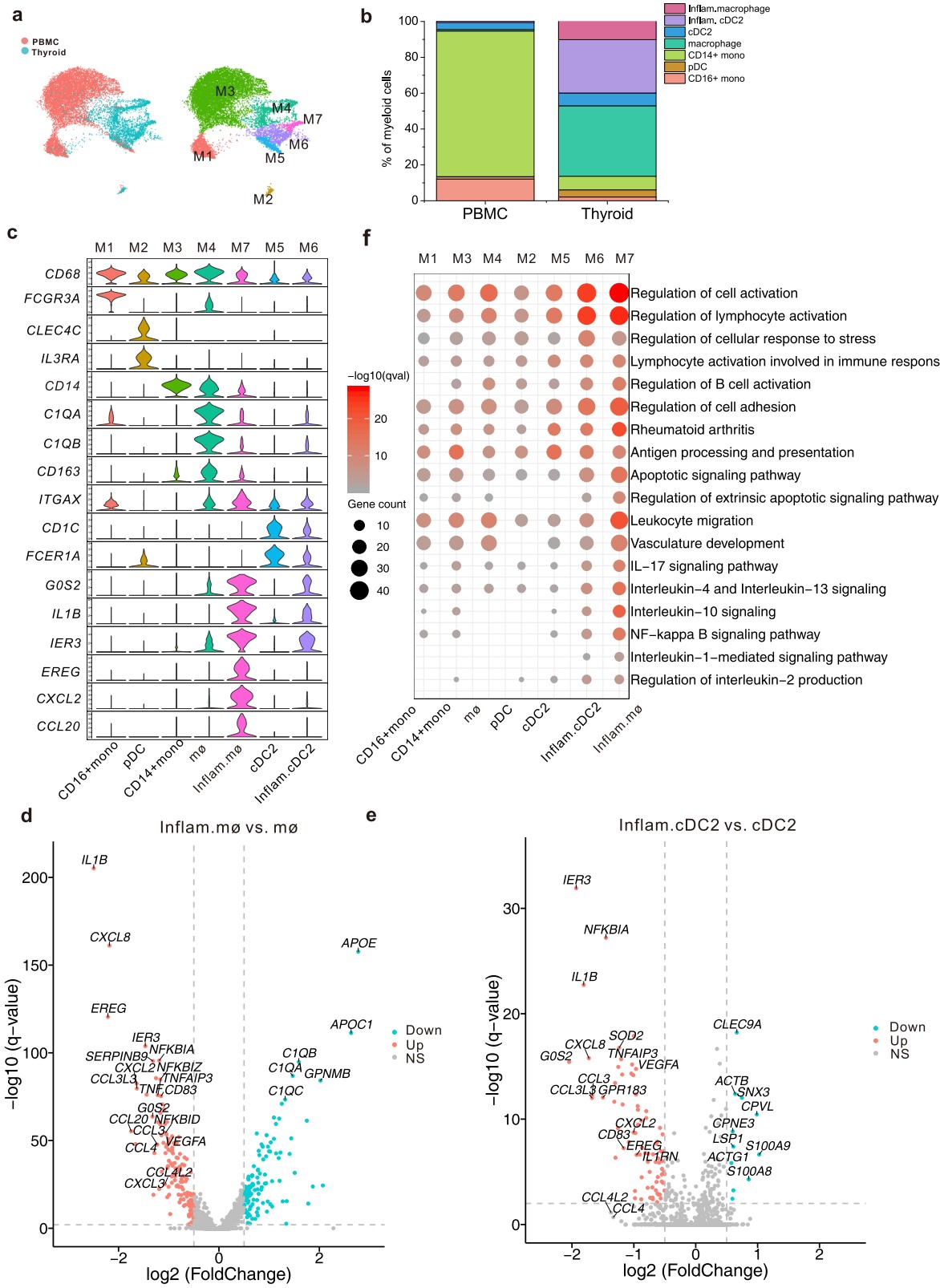

defined M6 and M7 as inflammatory cDC2 (inflam.cDC2) and inflammatory macrophages (inflam. mø), respectively.

Since inflam.cDC and inflam.mø existed exclusively in thyroid tissues of HT patients (Fig. 8a, b), after correcting the donor effects by linear mixed effect (LME) methods, the differential gene expressions between inflam.mø and mø, inflam.cDC and cDC2 were studied. Our results showed that cytokines and chemokines including *IL1B*, *TNFAIP3*, *NFKBIA*, *CCL3L3*, *CCL3*, *CXCL2* and *CXCL8* were more abundantly expressed in both inflam.cDC and inflam.mø than in cDC2 or mø, with *IL-1B* showing the most

**Fig. 8 Comparative analysis revealed possible pathogenic roles of thyroid-specific DC and macrophage subgroups in HT patients. a** UMAP of merged myeloid cells, colored by source of PBMC or thyroid tissues of HT patients (left) and cluster identity (right). **b** Total myeloid cell subgroup proportions in PBMCs and thyroid tissues of HT patients. **c** Violin plots showing the expression levels of representative marker genes across myeloid cells clusters. The y axis shows the normalized read count. **d** Volcano plot showing the differentially expressed genes between subgroup M4 and M7. Hypothesis tests for linear mixed effect models were two-sided with the Bonferroni correction. The x axis indicates log2 (FoldChange) of gene expression in M4 (macrophages) compared to M7 (inflammatory macrophages). Genes with $|log2FC| \geq 0.5$ and $-log10$ (q value) > 2 were colored as red (upregulated in M7) or blue (downregulated in M7). NS: no significance **e** Volcano plot showing the differential expressed genes between subgroups M5 and M6. Hypothesis tests for linear mixed effect models were two-sided with the Bonferroni correction. The x axis indicates log2 (FoldChange) of gene expression in M5 (cDC2) compared to M6 (inflammatory cDC2). Genes with $|log2FC| \geq 0.5$ and $-log10(q$ value) $\geq 2$ were colored as red (upregulated in M6) or blue (downregulated in M6). NS no significance. **f** Selected Gene Ontology (GO) terms and KEGG Pathways of myeloid cell subgroups. The statistical significance was tested by two-sided hypergeometric test and adjusted by Benjamini–Hochberg correction. Bubble color represents q values, and bubble size represents the enriched gene counts for the term. Mono monocytes, mø macrophages, inflam inflammatory.

significant difference (Fig. 8d, e, Supplementary Data 8). IL-1β was able to enhance CD40L-mediated activation of DC[47] and in vivo proliferation of antigen-stimulated naive CD4$^+$ TCs[48], promote Th17 cell responses[49], and induce the expression of Fas in thyrocytes, which is an important participant in thyrocyte apoptosis[50]. Consistently, CD83, a costimulatory molecule and activation marker of DC[51], was also upregulated in inflam.cDC and inflam.mø (Fig. 8d, e), which was indicative of a more activated state of inflam.cDC2. Indeed, GO enrichment analysis revealed several inflammatory pathways were activated in both inflam.cDC and inflam.mø, including IL-10, IL-4, IL-13, NF-κB, and IL-17 signaling pathways (Fig. 8f). IL-4 are critical for Th2 differentiation. Our results suggested inflam.cDC and inflam.mø could play essential roles in the aberrant Th2 and Th17 responses in HT pathogenesis. In addition, genes associated with lymphocyte activation, leukocyte migration, cell adhesion, apoptotic signaling pathway, were also enriched in inflam.cDC and inflam.mø (Fig. 8f), which was consistent with the proapoptotic effect of IL-1β and elevated chemokine expression levels in inflam.cDC and inflam.mø. Notably, the expressions of *CXCL8*, *VEGFA*, and *EREG* (a ligand for EGFR) were also significantly increased in inflam.cDC and inflam.mø compared with cDC2 and mø, respectively (Fig. 8d, e). CXCL8 has been shown to promote angiogenesis by upregulating VEGFA[52], which is well known in regulating angiogenesis. This was consistent with cellphoneDB data, which show growth factors such as VEGFA secreted by DCs and Macrophages could interact with their receptors expressed by ECs and lymphatic ECs (KDR and FLT1, respectively) (Fig. 3a). cellphoneDB analysis also showed *AREG* (expressed by DC and Mac) and *GRN* (expressed by DC and NK cells) may interact with *EGFR* (expressed by fibroblasts and myofibroblasts), contributing to the growth of these stromal cells (Fig. 3a). These data suggest that inflammatory macrophages and DCs exclusively exist in the thyroid of HT patients and could play crucial roles in the abnormal immune response, lymphocyte infiltration, and thyrocyte apoptosis observed in the pathogenesis of HT, mainly through the production of inflammatory mediators and angiogenesis-related factors.

## Discussion

A typical pathological feature of HT is the presence of lymphocytic infiltrates, and even TLOs, in thyroid tissues. Many cytokines and chemokines derived from those infiltrated lymphocytes have been shown to play important roles in the pathogenesis of HT[16]. However, the mechanisms underlying the infiltration of immune cells in thyroid tissues of patients with HT are poorly understood. In the present study, we found that a novel subset of fibroblasts (F1) in thyroid tissues of HT patients abundantly expressed the lymphoid chemokines CCL19 and CCL21 (Fig. 2b), as well as BC survival-related gene *TNFSF13B* (*BAFF*) and adhesive molecular *VCAM1* (Fig. 2b). Previous studies have

shown that fibroblastic reticular cells (FRCs), which are specialized stromal cells in the TC zone of germinal centers and abundantly express CCL19 and CCL21, are important stromal organizer cells that play an essential roles in the development of SLOs[53–55]. Moreover, aberrant overexpression of CCL21 in pancreas and thyroid tissues of transgenic mice was reported to induce the formation of TLOs in the pancreas and thyroid glands[54,56]. A study demonstrated that in normal lymphoid node (LN) tissues, *CCL21* mRNA was not detected in the HEVs, but the CCL21 protein could be observed by immunostaining, suggesting CCL21 produced by other LN stromal cells can be transcytosed across HEVs and thereby influence lymphocyte recruitment from the blood[57]. In RA, another autoimmune disease with the formation of TLOs as HT, CCL21 was abundantly expressed in synovial tissues[58,59]. However, the exact cell types and microarchitecture of CCL19 or CCL21 expressing cells in TLOs are not clear. Interestingly, we found many CCL21$^+$ fibroblasts were distributed in the TC zones of TLOs in thyroid tissues from HT patients (Fig. 5d–f), suggesting that CCL21$^+$ fibroblasts (F1) in TC zones of TLOs could be a counterpart of lymphoid tissue organizer cells, to promote lymphocytes recruitment and the formation of TLOs in thyroid tissues of HT patients.

Previous studies have reported that HEVs are the main entrances for lymphocytes to migrate from blood to lymphoid tissues during the development of SLOs, in which a complex set of steps is necessary, with the ECs playing pivotal roles in controlling their entry. In the present study, we found ACKR1$^+$ ECs (EC2) were distributed in the small venules in the thyroid tissues of patients with HT (Fig. 4a–c); however, not all the small venules with ACKR1$^+$ ECs (EC2) were HEVs. Although all ECs in HEVs highly expressed the ACKR1, only part of them were surrounded by myofibroblasts (double-positive for ACTA2 and CCL21, Fig. 5a), which were distributed on the adventitia of HEVs (Fig. 4e) or CCL21$^+$ fibroblasts (F1, Fig. 5a) that appeared abundantly around lymphocyte aggregates in thyroid tissues of HT patients (Figs. 4a, d, e and 5b). Notably, we also found that the number of ACKR1$^+$ vessels were dramatically increased in the thyroid tissues of HT patients compared to non-HT controls. Previous studies reported that ACKR1 is a receptor for multiple chemokines such as CCL2, CCL5, CXCL5, and CXCL8. Besides its well-defined role in red blood cells, ACKR1 was shown to be expressed on the inflamed synovial endothelium in RA patients[60], and it was essential for the recruitment of neutrophils in a multicellular model of RA synovium mediated by interaction with CXCL5[61]. Moreover, ACKR1 expression in ECs was reported to increase the efficiency of leukocyte recruitment, probably by transporting chemokines for presentation to circulating leukocytes[60,62]. Consistently, in migration studies in vitro, we proved that ACKR1 expressed by ECs facilitated the trans-endothelial migration of lymphocytes under chemotaxis of ACKR1 ligands CCL2 or CCL5 (Fig. 4f). Our scRNA-seq data

showed CCL2 was expressed by DCs, macrophages, especially myofibroblasts, and the F1 fibroblast (Fig. 2a, b, Supplementary Fig. 3D), while CCL5 was expressed by immune cells (especially CD8+ TCs and NK cells) of thyroid tissues (Supplementary Fig. 3D). This suggests ACKR1 may contribute to the infiltration of immune cells from blood through transferring chemokines secreted by thyroidal stromal cells and immune cells across HEV ECs. Together with other previous reports, it is tempting to presume that when autoimmune tolerance is disturbed in HT, myofibroblasts (AC2) or CCL21+ fibroblasts (F1), which surround the HEVs, promote lymphocyte recruitment from blood by secreting CCL19 and CCL21, which can bind to its receptor CCR7 expressed on TCs, BCs, monocytes, or DCs (Fig. 9). The recruited lymphocytes are further attracted to sites of thyroid tissue in HT patients by the high concentration of CCL21 secreted from CCL21+ fibroblasts (F1), contributing to the formation of TLOs in the region of thyroid tissue with aggregation of F1 cells as well as the infiltration of lymphocytes to the area with distribution of scattered F1 cells in HT patients (Fig. 9). Moreover, a number of novel HEVs with high ACKR1 expression on its ECs are generated in thyroid tissues of HT patients and facilitate the recruitment of lymphocytes by translocating the chemokines produced by infiltrated immune cells and thyroidal stromal cells such as CCL2 and CCL5, which further exacerbate the infiltration of immune cells in the thyroid tissue of HT patients (Fig. 9). However, more comprehensive studies will be needed to investigate whether CCL21 expression in myofibroblasts or fibroblasts alone is able to induce TLOs in the thyroid, and whether CCL21 contributes to the genesis of HEVs, as previous studies also identified CCL21 as a mediator of angiogenesis in RA[63].

Another important finding of our study is the identification of inflam. mø and inflam.cDC2 that specifically exist in thyroid tissues but not PBMCs of HT patients. DCs are well known to provoke organ-specific autoimmune diseases. Identification of DCs in the thyroid tissue of AITD patients and animal models has strongly argued for DC involvement in the initiation of autoimmunity[64,65]. Interestingly, both inflam.cDC2 and inflam.mø highly expressed

inflammatory chemokines and cytokines such as IL-1β. IL-1β is known to stimulate IL-6 secretion by thyrocytes, modify epithelium integrity, and inhibit thyrocyte growth[66,67]. Moreover, IL-1β was reported to induce FAS expression on TFCs and mediate TFC apoptosis through FAS-FASL signaling[50]. IL-1β has also been reported to be increased in thyroid tissues of patients with HT compared to those with GD[68], another AITD characterized by lymphocyte infiltration but no thyrocyte destruction. These data suggest thyroidal-specific inflammatory cDCs and macrophages in thyroid tissues in HT could play critical roles in thyrocyte destruction through IL-1β signaling.

To our surprise, though we also found MHC II was highly expressed in TFCs of HT patients, as reported in previous publications[69], the expression profile of TFCs was homogenous without the inflammatory phenotype, which is different from the previous notion that thyrocytes secret chemokines and cytokines after stimulation by IL-1β and IFN-γ, which are abundantly produced in thyroid tissues of HT patients. This may be partially explained by the fact that few TFCs, or only "healthy" TFCs, were detected in this study due to the fragility of thyrocytes that are prone to undergo apoptosis and the use of a Dead Cell Removal Kit in single-cell isolation.

Collectively, this study constructs the transcriptional landscape of thyroid tissues and PBMCs in HT patients and provides evidence for the thyroid as a main producer of autoantibodies. In addition, our study shows that specific stromal cells such as ACKR1+ ECs in HEVs, CCL21+ myofibroblasts, and fibroblasts appear in the thyroid of HT patients and play important roles in lymphocyte recruitment and TLOs formation, providing key insight into the mechanisms underlying the pathogenesis of HT. Moreover, the identification of thyroid-specific inflammatory DCs and macrophages indicates these chemokine-/cytokine-expressing cells could play potential roles in thyrocyte apoptosis and Th17 cell-mediated inflammation in the thyroid. Our data advance our understanding of the possible mechanisms underlying the pathogenesis of HT and could inspire new strategies for the treatment of HT.

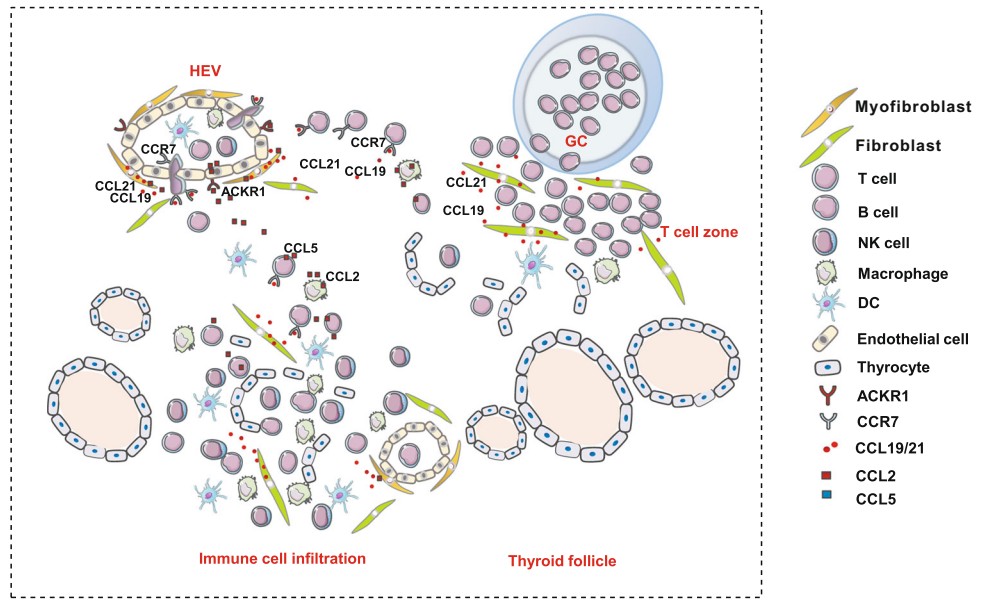

**Fig. 9 Hypothesized model of the roles stromal cells play in immune infiltration in HT.** In the thyroid tissues of HT patients, blood immune cells migrate through HEVs under the chemotaxis of CCL19 or CCL21 secreted by CCL21+ myofibroblasts and CCL21+ fibroblasts distributed in the adventitia of HEVs and around HEVs, respectively. Meanwhile, ACKR1+ endothelial cells facilitate the migration by translocating CCL2 and CCL5 secreted by CCL21+ myofibroblast/fibroblasts and immune cells to the lumen of HEVs. Then these infiltrated immune cells are attracted to the sites with scattered CCL21+ fibroblasts. In some regions of the thyroid, where TLOs with germinal centers are generated, CCL21+ fibroblasts are distributed in T cell zones and may play a role in structure organization and T cell signaling.

## Methods

**Study subjects**. The study was performed in accordance with protocols approved by the ethics committee of Shanghai Ninth People's Hospital. Patients were all recruited from the Shanghai Ninth People's Hospital. Five HT patients with thyroid neoplasms requiring thyroidectomy were included for scRNA-seq. Sixteen HT patients and 50 non-HT patients with thyroid neoplasms requiring thyroidectomy were included for bulk RNA-seq. All patients provided written informed consent for collecting their extra tissue after thyroidectomy and peripheral blood for the current research. HT was diagnosed by thyroid histopathological examination characterized by a diffuse lymphocytic infiltration and destruction of thyroid follicle cells or combined with elevated TGAb or TPOAb. The patients without HT were diagnosed if the individual has no lymphocytic infiltration and destruction of thyroid follicle cells on the thyroid histopathological examination. Characteristics of patients for single-cell RNA sequencing are summarized in Supplementary Table 1. Characteristics of patients for bulk RNA-seq are summarized in supplementary data 4.

**Single cells separation of thyroid gland and PBMCs**. Primary thyrocytes were isolated using a procedure described by Morgan et al. with minor modifications. Briefly, normal thyroid tissues were obtained from patients undergoing partial or total thyroidectomy for thyroid neoplasms. Thyroid tissue was kept on ice, minced into small pieces (approximately 1 mm$^3$) in HBSS, and then transferred to a 15-ml tube and washed at least twice with HBSS containing 1% FBS and 2 mM EDTA. Tissue pieces were then digested with HBSS containing 5 mg/ml collagenase II (Sigma-Aldrich, St. Louis, MO), 0.1 mg/ml dispase (Sangon Biotech, Shanghai, China), and 10% fetal bovine serum (FBS) (Gibco, USA), with constant shaking for 30 min at 37 °C in a water bath shaker until a suspension of isolated cells was obtained. After digestion, cells were filtered through a 70 μm cell strainer (352350, Falcon, USA) and centrifuged at $300 \times g$ for 5 min. The supernatant was discarded, and precipitates were resuspended with red blood cell lysis buffer for 2 min and then washed twice with HBSS containing 1% FBS and 2 mM EDTA to remove erythrocytes. Cell suspension was filtered through a 30 μm filter (Miltenyi Biotec, Bielefeld, Germany) to remove cell clumps thoroughly. After dead cells were removed by using the Dead Cell Removal Kit (Miltenyi Biotec, Bielefeld, Germany), cell viability was higher than 90% as assessed by trypan blue staining. PBMCs were isolated from 5 ml of anticoagulant EDTA-K2 peripheral blood by density gradient separation methods using lymphoprep (Stem Cell, Vancouver, Canada). Blood for the isolation of PBMCs used for scRNA-seq was taken just before operation.

**Single-cell RNA sequencing**. We performed single-cell RNA sequencing of all cells in the digested thyroid tissue and individual matched PBMCs. Cells were resuspended at final concentrations of 800–1000 cells/μl depending on the samples. Cellular suspensions of each sample were loaded on a GemCode Single-Cell Instrument (10x Genomics, Pleasanton, CA, USA) separately to generate single-cell Gel Bead-in-Emulsion). Single-cell RNA-Seq libraries of each sample were prepared using the Chromium Single-cell 3′ library and Gel Bead Kit v2.1 or v3.1. Each sample was processed separately to generate a library with a unique barcode for following sequencing. The single-cell library of each patient was sequenced in different lane of Illumina Nova S6000.

**Data processing and quality control**. Sequences obtained using the 10x Genomics single-cell RNA-seq platform were demultiplexed and mapped to the GRCh38 human transcriptome using Cell Ranger V3.0.1. The output of Cell Ranger is a raw count matrix for each sample, which records the unique molecular identifier (UMI) counts per gene and associated cell barcodes. Low-quality cells and genes were removed using Seurat V3.1.1 (http://satijalab.org/seurat/). Genes expressed in less than three cells were removed. Cells that had over 6000 unique feature counts were removed to exclude the cell doublets or multiplets, which may exhibit an aberrantly high gene count. Cells that had unique feature counts less than 200 were removed to exclude low-quality cells or empty droplets, which may often have very few genes. Cells having >25% mitochondrial counts and cells expressing high levels of hemoglobin-encoding genes were also discarded.

Computational inference of doublets was performed by the Scrublet package[70]. The expected doublet rate was set to 0.07. Doublets identified in each sample individually were excluded from the following analyses.

**Normalization, batch effect correction, clustering, and dimensional reduction in analyzing the scRNA-seq data of thyroid tissues**. Data normalization, cell clustering, and dimensional reduction were performed with the Seurat package. Expression data were normalized using the "NormalizeData" function with default settings. The effects of variation in sequencing depth were regressed out by including "nUMI" as a parameter in the "ScaleData" function. Two thousand highly variable genes (HVGs) were chosen from the corrected expression matrix and then centered and scaled using the "FindVariableGenes" function. These variable genes were used as input for PCA using the "RunPCA" function. The top 20 principal components were used for further analysis. Batch effects were corrected by "RunHarmony" function in the harmony package[71]. The sample identity (batch) was set as the covariate for correction and the corresponding theta parameter was set to 2.

Significant clusters were identified using the "FindClusters" function in the Seurat package, which uses a shared nearest neighbor modularity. The clustered cells were then projected onto a two-dimensional space using the "RunUMAP" function. For a secondary cluster analysis of each cell lineage, cells in clusters of interest were identified and extracted using the "SubsetData" function. Clustering and dimensional reduction of UMAP were repeated as described above.

**Identification of marker genes and differential expression genes for cell clusters of thyroid tissues**. LME methods were used to control high type I error rates when comparing cells[72]. The marker score of each gene for each cell group was computed based on the Jensen-Shannon divergence[73]. The marker significance test was conducted using LME models by R package lme4 (v1.1.27). Regression models were tested for cluster ID as a fixed effect and donor as a random effect. F statistics and $p$ values based on Satterthwaite's approximations were calculated with LmerTest package in R. $P$ values were adjusted by the Bonferroni method to generate $q$ values. The top 200 markers by marker score meeting a $q$ value < 0.01 were used for the following analyses.

Differentially expressed features between pairs of clusters were only tested on the genes detected in greater than 10% of cells in either of the two populations. The significance test was conducted using LME models by R package lme4 (v1.1.27). Regression models were tested for cluster ID as a fixed effect and donor as a random effect. F statistics and p-values based on Satterthwaite's approximations were calculated with LmerTest (v3.1.3) package in R. P-values were adjusted by the Bonferroni method to generate $q$ values. The top 200 markers by q value meeting a $q$ value < 0.01 were used for the following analyses.

**Integration of thyroidal immune cells and PBMCs**. Based on the differential expression of known and novel marker genes, 5 of 10 clusters were identified as immune cells: TCs, BCs, MCs, PCs, and PrCs. The raw data of these immune cells were extracted from thyroid datasets and merged with PBMC raw data using the "merge" function in the Seurat package. The normalization, correction of batch effects, clustering, and dimensional reduction were performed as described in the single-cell analysis of thyroid tissues.

LME methods were used to control the influence of the heterogeneity between thyroidal immune cells and PBMCs in identifying marker genes for each cell cluster and differentially expressed genes between paired cell clusters. For marker gene identification, the score of each marker was also estimated based on the Jensen-Shannon divergence[73]. Then, the marker significance test was conducted using LME models by R package lme4 (v1.1.27). Regression models were tested by including cluster ID as a fixed effect. Sample identity and tissue origin were included as random effects. F statistics and $p$ values based on Satterthwaite's approximations were calculated with LmerTest package in R. For identifying differentially expressed genes between two cell clusters, we also used the LME methods with the same parameters. Differential expression was only tested on the genes detected in more than 10% of cells in either of the two populations. The criterions of selecting top marker genes and top differentially expressed genes were the same as in analyzing the data of thyroid tissues.

**HE and Polychromatic immunofluorescence staining of thyroid sections**. Polychromatic immunofluorescence staining was performed by using the four-color multiple fluorescent immunohistochemical staining kit (abs50012, Absin, Shanghai, China) based on the Tyramide signal amplification (TSA) technique following the manufacturer's manual. Briefly, 5-μm tissue sections from formalin-fixed, paraffin-wax-embedded human thyroid tissues were dewaxed with xylene, cleared in 100% ethanol, and rehydrated through gradients of ethanol. Sections were subjected to microwave-induced antigen retrieval in sodium citrate buffer (PH = 6.0) and endogenous peroxidase blocking in 0.3% hydrogen peroxide in methanol. Then, sections were washed in PBST and blocked with 5% goat serum in PBS for 10 min, incubated with primary antibody for 1 h at room temperature, and slides were washed in PBST. Horseradish peroxidase-labeled goat anti-rabbit/mouse secondary antibody was used and was developed with fluorescent dye diluted by the signal amplification reagent provided with the kit. For multiple fluorescent staining, sections were processed starting from the antigen retrieval step to remove binding antibodies, and then they were incubated with another primary antibody. This was repeated until all antigens were stained. Lastly, sections were counterstained with DAPI and mounted in glycerol and gelatin mounting medium. Antibody information is listed in Supplementary Table 2. Antibodies were validated for IHC in Human tissues as noted on the manufacturer's website as cited in supplementary Table 2. Additional validation was done by the use of negative control samples in each experiment. Tissue sections were imaged using a Nikon A1 scanning confocal microscope. The confocal images were captured with a 20× or 10× objective and the image data was collected using NIS Elements (Nicon, V4.50.00). Imaris 9.0.1 was used for the analysis of image data.

**RNAscope detection of *CCL21***. The RNAscope assay was performed on thyroid sections using RNAscope Multiplex Fluorescent Reagent Kit v2 [323100, Advanced Cell Diagnostics (ACD), Hayward, CA]. Briefly, tissue sections were deparaffinized with xylene and 100% ethanol and incubated with hydrogen peroxidase for 10 min at room temperature, target retrieval regent for 30 min at 98–102 °C, and protease

plus (Pretreatment kit 322381, ACD) for 15 min in the HybEZ hybridization oven (ACD) at 40 °C. The slides were then hybridized with the probe for Hs-CCL21 (310061) or 3-plex Positive Control Probe-Hs (320861) or 3-plex Negative Control Probe (320871) in the oven at 40 °C for 2 h. After hybridization, slides were subjected to signal amplification by incubation with AMP1, AMP2, AMP3, and C1-HRP (in order) using the detection Kit, and the hybridization signal was detected using Opal 570 (FP1488001KT, Akoya Biosciences, USA) (1:1500). After blocking with the HRP-block in the detection kit, slides were subjected to polychromatic immunofluorescence staining using the four-color multiple fluorescent immunohistochemical staining kit as described in the above section for protein targets.

**Cell culture and migration analysis**. C2C12 mouse myoblasts cell line was obtained from ATCC, and T8.1, a murine hybridoma TC line[74,75], was obtained from the Chinese Academy of Science in Shanghai. The transmigration assay was conducted on 24-well plates with 8 μm transwell inserts (MCEP24H48, Millipore, USA) in triplicate. T8.1 and C2C12 were cultured with DMEM (Yuanpei, Shanghai, China) containing 10% FBS (Gibico, USA). The mouse ccl21 coding sequence was cloned into PCDNANA3.1 and transfected into C2C12 at 80% confluence with the vector as control. Transfection media were replaced by DMEM containing 0.15% BSA 6 h after transfection. After 24 h, the media of ccl21-PCDNA3.1 or vector-transfected cells were collected and centrifuged to remove cell fractions. Then, 900 μl media was added to the 24-well plate with transwell inserts, and $2 \times 10^5$ T8.1 cells (200 μl) were seeded in the upper chamber. After incubation at 37 °C for 20 h, cells that migrated to the lower chamber were collected and counted by a cell counter after trypan blue staining.

The trans-endothelial migration experiment was performed as previously reported[76]. HUVECs were obtained from FuHeng Cell Center (FH1122, Shanghai, China). HUEVC cells of 4–8 passages were cultured with specific endothelial culture media (ECM) (ScienCell, San Diego, USA). Then, 24 h after transfection with the pcDNA3.1 vector encoding Human ACKR1 or the empty vector, HUEVCs were grown on transwell inserts until they were confluent. Confluence of the monolayers was verified by the addition of red blood cells. Then, monolayers were stimulated by 2 ng/ml IL-1β to promote the expression of adhesive molecules for 12 h and were washed twice. Next, 900 μl media (RPIM1640 containing 0.15% BSA) containing Human CCL2 (10 ng/ml) and CCL5 (10 ng/ml) was added to the lower chamber, and $4 \times 10^5$ PBMCs in the media with no chemokines were added to the inserts and were allowed to migrate across the monolayers for 20 h. PBMCs that had migrated were counted by a cell counter after trypan blue staining.

**Cell–cell interactions analysis**. The expression matrix together with cell type metadata was used as input to cellphoneDB[26,27]. Genes expressed by less than 10% of cells per cell cluster were removed. Cell–cell interactions were predicted against a null distribution using a known receptor-ligand database. Number of iterations for the statistical analysis was set to 10,000. Significant cell–cell interactions were selected with $p$ value < 0.01 in our cellphoneDB analysis. The interactions were shown by graphs drawn with the R "igraph" package (v1.2.5). Cell types were nodes and interactions were edges. Cell type size was proportional to the total number of interactions.

**Bulk RNAseq analysis**. Sixty-six thyroid tissues were collected from 16 patients with HT and 50 patients without HT (non-HT). Total RNA was isolated from tissues using Trizol reagent (Invitrogen, USA) and quantitated by Qubit 2.0 Fluorometer (Thermo Fisher Scientific, USA). All samples had an RNA integrity number ≥7, indicating good quality. RNA was reverse-transcribed, labeled with beads, amplified, and the library was constructed in three different batches using the Illumina KAPA Stranded mRNA-Seq Kit (Kapabiosystems, USA). Within a total number of 66 samples, adapter and low-quality reads of raw data were first trimmed using "Trim_galore" with a threshold Phred score less than 30 and total minimum length less than 35 pb for each read. Clean reads were then aligned to hg38 using "Hisat2". "Samtools" and "Picard" were used for sorting and removing duplicated reads. FeatureCounts were used for read counts. Gene expression difference between HT and non-HT samples was firstly analyzed by applying "DESeq2" package on raw read counts. The gene expression levels were extracted from "DESeq2" results. To control for the effects of covariates, we conducted LME models for the significance test by R package "lme4". Sample group (HT or non-HT), sex, and age were included as fixed effects. Library preparation batches were included as random effects. F statistics and p-values based on Satterthwaite's approximations were calculated with "LmerTest" package in R. $P$-values were adjusted by the Benjamini–Hochberg method to generate q-values. The differentially expressed genes meeting absolute log2 fold-change > 0.25 and $q$ value < 0.01 were sorted by $q$ value and used for the following analyses.

**Gene set variation analysis (GSVA) and pathway enrichment analysis**. The marker genes with high specificity values for each subcluster detected in scRNA-seq were selected as the gene sets in the GSVA analysis for Bulk RNAseq. The expression matrix extracted from the Bulk RNAseq analysis was compared to gene sets by calculating average enrichment scores for each cluster using the "gsva" function in the GSVA (V1.34.0) package. Significantly different GSVA scores and gene expression between HT and non-HT patients were determined by two-sided

Wilcoxon Rank Sum tests with Benjamini–Hochberg corrected $p$ value < 0.05 and are represented as heatmaps by rows.

Pathway enrichment of marker genes was analyzed using the Metascape tool (http://metascape.org) with default parameters. Functional enrichment analysis was performed based on the following genomics sources: KEGG Pathway, Gene Ontology, Reactome Gene Sets, and Canonical Pathways. The statistical significance was tested by hypergeometric test and adjusted by Benjamini–Hochberg p value correction algorithm. Terms with q value < 0.01, minimum count 3, and enrichment factor >1.5 were collected and grouped into clusters based on their membership similarities. Kappa scores were used as the similarity metric, and sub-trees with a similarity >0.3 were considered a cluster. The most statistically significant term within a cluster was chosen to represent the cluster.

**Statistics and reproducibility**. Additional analysis or visualization was performed using the R packages ggplot2 (v3.2.1) and Seurat in R v3.6.1. SPSS 22.0 software was used for Student's t test. Originpro 8.5.1 was used for histograms plotting. HE staining, polychromatic immunofluorescence staining and RNAscope detections were confirmed in at least three biological replicates with consistent results.

**Reporting summary**. Further information on research design is available in the Nature Research Reporting Summary linked to this article.

## Data availability
Single-cell sequencing data and bulk RNA sequencing data generated in this study have been deposited in the Genome Sequence Archive in National Genomics Data Center, China National Center for Bioinformation/Beijing Institute of Genomics, Chinese Academy of Sciences, GSA-Human access number: GSA-Human access number: HRA001684. Known receptor-ligand interactions can be downloaded from cellphoneDB. Human reference (GRCh38) dataset required for Cell Ranger can be downloaded at 10x Genomics. All other data generated in this study are presented in the Supplementary data or source data provided with this paper. Source data are provided with this paper.

## Code availability
All the codes are available under accession code 4575509 (https://zenodo.org/).

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

## Acknowledgements
We thank all members of the core laboratory at the medical center of clinical research at Shanghai Ninth People's Hospital for their valuable support and discussion. The work was supported in part by the National Natural Science Foundation of China (82070816, S.X.Z, 81870537, S.X.Z, 81800749, X.P.Y, 81770786, H.D.S), National Key R&D Program of China (2017YFC1001801, H.D.S), and Shanghai Municipal Education Commission Two-hundred Talent (20161318, S.X.Z). We acknowledge Servier Medical Art (https://smart.servier.com/) for providing image components of Figs. 1a, 3b, and 9, licensed under a Creative Commons Attribution 3.0 Generic License (https://creativecommons.org/licenses/by/3.0/).

## Author contributions
Q.Y.Z. designed and performed experiments, and wrote the manuscript. X.P.Y., Z.Z. performed experiments, evaluated, and interpreted the data. J.N.L. and S.Y.S. performed scRNA-seq analysis and data interpretation. Y.F., R.J.Z., Z.W., C.F.Z., L.L., W.L., R.L., and S.Y.L. helped with the experiments. S.X.Z., C.F.Z. supervised the study, and revised the manuscript. H.D.S. conceived the idea, revised the manuscript, coordinated and supervised the studies.

## Competing interests
The authors declare no competing interests.
