## [Peer Review File · Nature Communications]

Lymphocyte Infiltration and Thyrocyte Destruction is Driven by Stromal and Immune Cell Components in Hashimoto's ThyroiditisREVIEWER COMMENTS

Reviewer #1 (Remarks to the Author):

In the manuscript entitled "Thyroidal stromal and immune cell subsets promote lymphocyte infiltration and thyrocyte destruction in Hashimoto's thyroiditis", the authors performed single-cell RNA-seq from thyroid and PBMCs of HT patients to characterize the cellular subsets present in the thyroid. The authors complement their findings with bulk RNA-seq of para-tumor samples of HT versus non-HT patients, as well as with imaging and cell migration assays.

I found insightful learning about the cell subsets present in thyroid of HT patients, what chemokines and receptors they express, and how they might contribute to disease pathogenesis. However, in several parts of the manuscript I think the authors over-claim their findings, and I have important concerns with the data analyses.

Major concerns:

- What was the study design for the single-cell RNA-seq experiment? Was each sample processed in a separate lane of 10X? When the authors used Harmony, what parameters were used? What was the variable for batch correction?
- For finding the markers per cell cluster, what parameters were used in those functions? Did the authors account for donor effects? I would suggest fitting a mixed effects linear regression model with donor as a random effect and see how it compares to the current list of differentially expressed genes.
- When applying the CellPhoneDB analysis and filtered for $P < 0.01$. Did the authors correct for multiple hypothesis testing? How would the results look if a more stringent threshold is applied accordingly?
- For the bulk RNA-seq analyses, the study design and the parameters used for most methods are missing. How many library preparation batches are there and how do HT and non-HT patients distribute across them? For differential expression analysis, did the authors control for covariates that might segregate with HT vs Non-HT status (library preparation plate and clinical covariates)?

- In the section titled "Inflammatory DCs and macrophages presented exclusively in thyroid tissues of patients with HT and may play critical roles in the pathogenesis of HT"
 - o The title is misleading as it can make the reader think that the DCs and macrophages mentioned are exclusively in patients compared to controls.
 - o How did the authors integrate the PBMC and thyroid single-cell datasets? There is a clear separation in the UMAP by tissue, which suggests these differences could be largely driven by batch effects. The authors could apply harmony with batch correction by tissue of origin and donor. After this, the authors could perform clustering analysis again. The DC and macrophage cluster that appear "exclusive" for thyroid compared to PBMCs might not really be exclusive. For differential expression analysis between PBMCs DC2 and thyroid DC2, the authors should at least account for donor effects, and acknowledge that it can't be discarded that other batch effects (such as very different processing of thyroid samples compared to PBMCs) could be driving the observed differences.

- Can the imaging section be made more quantitative? It is often hard to see with certitude in the images what the authors claim in the text.
- In the sentence: "Interestingly, we found most of the CCL21-positive cells were distributed in T cell zones of ELS in thyroid tissues of HT patients (Fig. 4D)," The CCL21+ cells are clearly among immune cells, but lack of CD20 expression doesn't automatically mean T cell zone. Can the authors please explain how they arrive to that conclusion?
- In the discussion, "Notably, we also found that the number of HEVs were dramatically increased in the thyroid tissues of HT patients compared to non-HT controls" To my knowledge, the authors didn't actually show this.

Minor comments:

- In the sentence: "The infiltrated immune cells accounted for 65% to 82% of all the cells in thyroid tissues of HT patients, most of which were T cells, B cells, and plasma cells." Could these percentages be so high because of technical issues, in which stromal cells die more during the processing of the sample?
- "Interestingly, in addition to immune cell subgroups, the expression scores of the top markers for stromal cell subgroups EC2 and F1 identified by Monocle in the thyroid tissues of HT patients were significantly higher than those of non-HT controls, suggesting HEV EC2 endothelial cells and F1 inflammatory fibroblasts are increased in thyroid tissues of HT patients (Fig. S4B)." Can a P-value be provided for this? Visually, the difference doesn't look significant in Fig. S4B.
- "Consistently, further investigation of specific markers for EC2 and F1 showed that the expression levels of ACKR1, CCL19, CCL21, CXCL9, SELL, SELP, IL33, and MADCAM1 were significantly increased in thyroid tissues of HT patients (fig. S4C, table S4)." Similarly, what is the P-value for this differential expression of each of these genes? What is the fold change? CCL21, CXCL9 don't seem significantly differentially expressed.
- How does CD38 expression look in the subset that the authors annotated as PCs compared to other subsets? The expression levels of the marker the authors chose for PCs is not that high compared to other subsets.
- In the migration assays, what exactly are the error bars showing? Can the authors show the individual datapoints?

Reviewer #2 (Remarks to the Author):

The article "Thyroidal stromal and immune cell subsets promote lymphocytic infiltration and thyrocyte destruction in Hashimoto's thyroiditis" describes the results from the first scRNAseq in autoimmune thyroid disease. Results from scRNAseq and the accompanying bioinformatic analysis are complemented by some total RNAseq and immunohistopathology experiments. The work is well done and deserves publication.

Overall it is a bit disappointing that after generating so much data the amount of new information appears modest; the expression of a chemokine receptor (ACKR) that was missed in studies that go back 15 years and a finer dissection of fibroblasts, myofibroblasts and endothelial cells in the high endothelial venule.

The thyroids are clearly very heavily infiltrating and thyroid s cells loss due to sample preparation because the proportion seems too low.

The authors focus the analysis of these stromal cell because they seem to be crucial, as in canonical secondary lymphoid organs, for then generation of tertiary lymphoid tissue, and some of their result support their proposal, but do not demonstrate it.

Weak points

The lack of control normal thyroid tissue; it is fundamental to support the proposal that the stromal and capillary endothelial cells have acquired special function to generate / maintain autoimmunity, to demonstrate that they are different from the stroma of normal thyroid. Obtaining normal thyroid from nodule resection should be possible.

immunohistochemistry and immunofluorescence images are not top quality. Details will show better if in a frame and magnified inside the picture. The dynamics of cells can nor be directly be inferred from location (line 382), it should say suggest...

The difficulty in discerning the Th subtypes is not all surprising but Tfh and Tfr should perhaps be looked specifically for.

Minor

Most people refer to ectopic secondary lymphoid tissue as tertiary lymphoid organs (TLO); it may be a good idea they use this name

The paper language is in general clear, but abstract and discussion need a bit of editing

Non standard abbreviations such as info.mac and info.DC better avoided

Line 206, when the authors say "lymphocytic endothelial cells", they probably mean lymphatic vessels endothelial cells"

The use of T8.1 cells should be explained as it is not so common

Legend fig 1, there is no description for abbreviation PB, NaB, GCN (Germinal Center B cells?)
The article "Thyroidal stromal and immune cell subsets promote lymphocytic infiltration and thyrocyte destruction in Hashimoto's thyroiditis" describes the results from the first scRNAseq in autoimmune thyroid disease. Results from scRNAseq and the accompanying bioinformatic analysis are complemented by some total RNAseq and immunohistopathology experiments. The work is well done and deserves publication.

Overall it is a bit disappointing that after generating so much data the amount of new information appears modest; the expression of a chemokine receptor (ACKR) that was missed in studies that go back 15 years and a finer dissection of fibroblasts, myofibroblasts and endothelial cells in the high endothelial venule.

The thyroids are clearly very heavily infiltrating and thyroid s cells loss due to sample preparation because the proportion seems too low.

The authors focus the analysis of these stromal cell because they seem to be crucial, as in canonical secondary lymphoid organs, for then generation of tertiary lymphoid tissue, and some of their result support their proposal, but do not demonstrate it.

Weak points

The lack of control normal thyroid tissue; it is fundamental to support the proposal that the stromal and capillary endothelial cells have acquired special function to generate / maintain autoimmunity, to demonstrate that they are different from the stroma of normal thyroid. Obtaining normal thyroid from nodule resection should be possible.

immunohistochemistry and immunofluorescence images are not top quality. Details will show better if in a frame and magnified inside the picture. The dynamics of cells can nor be directly be inferred from location (line 382), it should say suggest...

The difficulty in discerning the Th subtypes is not all surprising but Tfh and Tfr should perhaps be looked specifically for.

Minor

Most people refer to ectopic secondary lymphoid tissue as tertiary lymphoid organs (TLO); it may be a good idea they use this name

The paper language is in general clear, but abstract and discussion need a bit of editing

Non standard abbreviations such as info.mac and info.DC better avoided

Line 206, when the authors say "lymphocytic endothelial cells", they probably mean lymphatic vessels endothelial cells"

The use of T8.1 cells should be explained as it is not so common

Legend fig 1, there is no description for abbreviation PB, NaB, GCN (Germinal Center B cells?)

Reviewer #3 (Remarks to the Author):

The manuscript's major claims are: 1) it is the first report that by using ScRNA-seq identified in the thyroid gland from patients suffering Hashimoto Thyroiditis (HT) the stromal cells that could promote lymphocytic infiltration. 2) identified the cells that will promote the formation of ectopic lymphoid tissue in the thyroid 3) The presence of inflammatory macrophages and dendritic cells in the thyroid. The authors contribute organizing and clustering groups of cells with markers and in this way they identified which cells could be responsible for stimulating the lymphocyte infiltration. Even though this is a tremendous and important contribution to identify the cells that bear the markers for inflammation or for recruiting immune cells, to my knowledge the authors have not been demonstrated that these cells are totally responsible for the pathogenesis of TH. The authors should be more cautious regarding their conclusion that they proved that these cells will facilitate lymphocyte traffic because even though they bear the markers like CCL21 the in vitro strategy is not enough to prove this. The correct approach will be that the authors should eliminate these cells and prove that there is not lymphocyte infiltration. Therefore I recommend changing the word "prove" in the abstract. The tables are missing in the manuscript and there are several phrases in the introduction that not have references and they should have them. The authors should present results of scRNA-seq regarding which type of antibodies are expression B cells in the thyroid and PBMC. This aspect it is important given that HT is characterized by antibodies against TPO or TG mainly and the question should be where are these cells, in the plasma or in the thyroid. The results of this manuscript are novel because it is the first work that uses this potent experimental strategy that is ScRNA-seq to identify the pattern of molecules that are expressing cells located in the thyroid gland of HT patients. I attached the manuscript where I added my comments. Therefore I recommend it for publication. This knowledge is relevant because it can be used for the diagnosis and treatment of HT. Moreover, it will also contribute to basic science to help in the characterization of cell types that invaded the thyroid gland and organized ectopic lymphoid structures based on the transcriptome. Yes, the work in this manuscript is convinced and it will be of interest to immunologists and endocrinologists. I attached the manuscript with my comments to improve it.

Reviewer #1 (Remarks to the Author):

In the manuscript entitled “Thyroidal stromal and immune cell subsets promote lymphocyte infiltration and thyrocyte destruction in Hashimoto’s thyroiditis”, the authors performed single-cell RNA-seq from thyroid and PBMCs of HT patients to characterize the cellular subsets present in the thyroid. The authors complement their findings with bulk RNA-seq of para-tumor samples of HT versus non-HT patients, as well as with imaging and cell migration assays.

I found insightful learning about the cell subsets present in thyroid of HT patients, what chemokines and receptors they express, and how they might contribute to disease pathogenesis. However, in several parts of the manuscript I think the authors over-claim their findings, and I have important concerns with the data analyses.

Major concerns:

- What was the study design for the single-cell RNA-seq experiment? Was each sample processed in a separate lane of 10X? When the authors used Harmony, what parameters were used? What was the variable for batch correction?

Response: We appreciate the reviewer’s comment since it touches on a very important information for our experiments. The thyroid tissues and PBMCs of 5 HT patients diagnosed by elevated TGAb or TPOAb combined with pathological assessment of thyroid tissue sections on HE stains were used for scRNA-seq. Primary single cells isolated from thyroid tissues or PBMCs of each patient were loaded on a GemCode Single-Cell Instrument separately to establish the expression library of single-cells. Each sample was processed separately to generate a library with a unique barcode for following sequencing. The single-cell library of each patient was sequenced in different lane of Illumina Nova S6000.

Batch effects were corrected by "RunHarmony" function in the harmony package both in the merging of the single-cell datasets from 4 thyroid tissues and the merging of the immune cell datasets which derived from 4 PBMCs and 4 thyroid tissues of HT patients. The sample identity (batch) was set as the covariate for correction and the corresponding theta parameter was set to 2. These detail information for our experiments have been added in the revised manuscript.

- For finding the markers per cell cluster, what parameters were used in those functions? Did the authors account for donor effects? I would suggest fitting a mixed effects linear

regression model with donor as a random effect and see how it compares to the current list of differentially expressed genes.

Response: Thank you for your comment. We used the Monocle3 methods to identify the markers from per cell cluster. In brief, key marker genes for per cell cluster were identified by the “top_markers” function in Monocle3 V0.2.0. The marker score of each gene for per cell cluster was computed based on the Jensen-Shannon divergence. The marker significance test was conducted using general linear model with a binomial family function and *P*-values were computed based on a likelihood ratio test between a full and simplified model of expression. *P*-values were adjusted by the Bonferroni method to generate *q*-values. The top 200 markers by marker score with a *q* value < 0.01 were used for the following analyses.

For the concern of the reviewer 1 about the influence of “donor effects” on the markers of per cell cluster, we used the linear mixed effect (LME) methods to correct the donor effect. In the LME method, the score of each marker was also estimated based on the Jensen-Shannon divergence as the Monocle3 method. The marker significance test was conducted using LME models by R package lme4. Regression models were tested including cluster ID as a fixed effect and sample identity as a random effect. *F* statistics and *p*-values based on Satterthwaite’s approximations were calculated with lmerTest package in R. *P*-values were adjusted by the Bonferroni method to generate *q*-values. The top 200 markers by marker score and meeting a *q* value < 0.01 were used for the following analyses.

In thyroid dataset, only 19 in top 200 marker genes of T cells identified by LME method are different with Monocle3 method as showed by Venn diagram. Moreover, both the order and gene names of the top 20 markers generated by the two methods are the same. For endothelial cells, the top 200 markers generated by the two methods are the same. An average of 96% markers identified by Monocle3 methods can be recovered by LME methods. Thus, we still use the result from Monocle3 methods for thyroid dataset in the revised manuscript.

- When applying the CellPhoneDB analysis and filtered for $P < 0.01$. Did the authors correct for multiple hypothesis testing? How would the results look if a more stringent threshold is applied accordingly?

Response: The suggestion of the reviewer about the correction of CellPhoneDB analysis by multiple hypothesis testing is very important. CellPhoneDB is a well-accepted method for inferring cell-cell communication from combined expression of multi-subunit ligand-receptor complexes in scRNA-seq data analysis (Efremova et al. Nat Protoc. 2020 Apr;15(4):1484-1506). In CellPhoneDB analysis, a *P* value indicates the likelihood of cell-type enrichment of a given receptor-ligand complex. As the documentation of CellPhoneDB, the cluster labels of all cells were randomly permuted (1,000 times by default) and the mean of the average receptor expression level in a cluster and the average ligand expression level in the interacting cluster was also determined. In this way, a null distribution for each ligand-receptor pair in each pairwise comparison between two cell types was generated. Then a *p*-value for the likelihood of cell-type enrichment of each ligand-receptor complex was obtained by calculating the proportion of the means which are as high as or higher than the actual mean. In other words, if the observed mean is in the top 5%, the interaction is assigned a *P* value of 0.05.

As the limitation of CellPhoneDB method in calculating *P* value, the *P* value will be assigned as zero if a *P* value is less than 0.001 when permuting 1,000 times. Thus, the huge number of tests from the big matrix of ligand-receptor interactions will cause *q*-value as either zero or one when performing multiple hypothesis test correction.

In this revised version, we improved the permutation by 10,000 times (the running time increased exponentially). Using the criterion of *P* value < 0.01, only 12,425 tests were significant in the total 385,246 tests. In the significant tests, 12,203 tests were assigned *P* value as zero. Thus, the *P* values from CellPhoneDB are unsuited for multiple hypothesis correction and are more suitable as indication of the likelihood of cell-type enrichment of each ligand-receptor complex.

- For the bulk RNA-seq analyses, the study design and the parameters used for most methods are missing. How many library preparation batches are there and how do HT and non-HT patients distribute across them? For differential expression analysis, did

the authors control for covariates that might segregate with HT vs Non-HT status (library preparation plate and clinical covariates)?

Response:

1. Thank you for your reminding. We have improved the method by adding the study design explanation and more analyses parameters in the revised manuscript as the following: “The para-tumor thyroid samples for bulk RNA-seq were from 16 HT patients and 50 non-HT controls. The patients with HT were diagnosed according to typical HT appearance on thyroid histopathological examination characterized as a diffuse lymphocytic infiltration and destruction of thyroid follicle cells. The patients without HT were diagnosed if the individual has no lymphocytic infiltration and destruction of thyroid follicle cells on the histopathological examination of the thyroid tissue.
2. The concern of the reviewer 1 about “How many library preparation batches are there and how do HT and non-HT patients distribute across them” have been described in details in the revised manuscript as the following: “The bulk RNA-seq library of the thyroid tissues from patients with or without HT were prepared in 3 different batches. The distribution of the patients with or without HT in the 3 different batches are showed in following histogram.”

3. Thank you for your good suggestion. For differential expression analysis, we did not control for covariates in the first version of our manuscript. In the revised version, we provided the clinical information of patients and controls. When projected in the tSNE space, we observed that HT samples were almost separated

from non-HT samples. However, the RNA-seq data for all samples with different covariates, such as batch information, sex and age, were almost mixed in the tSNE space. But after controlling for the covariates of age, sex, library batch by linear mixed effect (LME) models for the expression significance test, we got a more significant GO enrichment results.

Thus, we revised the results of bulk RNA-seq by replacing them with new data generated by LME models. We have also provided the sample information in supplementary table 5.

Old DESeq2 method

Linear mixed effect models

- In the section titled "Inflammatory DCs and macrophages presented exclusively in thyroid tissues of patients with HT and may play critical roles in the pathogenesis of HT"

-The title is misleading as it can make the reader think that the DCs and macrophages mentioned are exclusively in patients compared to controls.

Responses: Thank you for your nice suggestion. We changed the title to “Inflammatory DCs and macrophages presented in thyroid tissues but not in PBMCs in the patients with HT” in the revised manuscript.

How did the authors integrate the PBMC and thyroid single-cell datasets? There is a clear separation in the UMAP by tissue, which suggests these differences could be largely driven by batch effects. The authors could apply harmony with batch correction by tissue of origin and donor. After this, the authors could perform clustering analysis again. The DC and macrophage cluster that appear "exclusive" for thyroid compared to PBMCs might not really be exclusive. For differential expression analysis between PBMCs DC2 and thyroid DC2, the authors should at least account for donor effects, and acknowledge that it can't be discarded that other batch effects (such as very different processing of thyroid samples compared to PBMCs) could be driving the observed differences.

Responses:

The concern of the reviewer about “How did the authors integrate the PBMC and thyroid single-cell datasets” is important. In the revised manuscript, we described the

the method how we integrate the PBMC and thyroid single-cell datasets as following: we integrated the single-cell dataset of PBMC and thyroid immune cell clusters based on thyroid single-cell analysis. In brief, 5 of 10 clusters were identified as immune cells: T&NK cells (TC), B cells (BC), myeloid cells (MC), plasma cells (PC) and proliferative cells (PrC). The raw data of these immune cells were extracted from datasets of thyroid tissues and merged with PBMC raw data using the "merge" function in the Seurat package. The normalization, batch effect removing, clustering and dimensional reduction were performed as described in the single-cell analysis in thyroid tissues. These explanation for integrate the single-cell dataset of PBMC and thyroid immune cell clusters have been added in the revised manuscript.

The worry of the reviewer about “the differences of the immune cell cluster between thyroid tissues and PMBCs of HT patients could be largely driven by batch effects” are correct. In the first version of our manuscript, batch effects were corrected by "RunHarmony" function in the harmony package. In brief, the sample identity (a total of 8 samples) was set as the covariate for correction. The results after correcting batch effect is displayed in following pictures (A). As is showed in the picture, T/NK cells and B/Plasma cells from different origins (PBMC or thyroid) can be well integrated but myeloid cells were distributed differently (A). Here, according to the suggestion of the reviewer, we also tried to set both sample identity (a total of 8 samples) and tissue origin (PBMC or Thyroid) as the technical covariates for correction the batch effect (B). The results are similar as above. Thus, we believe that large differences do exist in the myeloid cells between that from PBMC and from thyroid tissues. In fact, monocytes in circulation are found to acquire huge functional and morphological changes when they entered tissues according to the local environments. DCs are also reported to go on functional changes such as being stimulated to activation after entering tissues. Thus, it was expected that the expression profile of PBMC myeloid cells is differed a lot from the thyroid myeloid cells.

For differential expression analysis (Revised Fig. 8E) between inflammatory cDC2 (M6, almost from the thyroid tissues) and cDC2 (M5, from both PBMCs and the thyroid tissues), we corrected donor effects following the suggestion of the reviewer (Revised

Fig. 8A, B). As a result, we observed similar expression differences as showed in the first version of our manuscript after correcting donor effects.

By the way, when we identified the marker genes of each cell subsets from the merged data of PBMC and the immune cells of thyroid tissues, we used the linear mixed effect (LME) methods to correct the donor effect according to the second suggestion of the reviewer 1. Sample identity and tissue origin were included as random effects. The result showed an average of 93% markers identified by Monocle3 methods can be recovered by LME methods. Lower proportion of recovered markers should indicate the influence of donor effects. Thus, we revised the results of marker identification and differential expression analysis using LME models for the merged immune cells (revised Fig. 8D-F).

(A) UMAP of the merged immune cells after removing batch effect (sample identity as the technical covariates for correction) colored by patient ID (left) or tissue origin (right). (B) UMAP of the merged immune cells after removing batch effect (sample identity and tissue origin as the technical covariates for correction) colored by patient ID (left) or tissue origin (right).

- Can the imaging section be made more quantitative? It is often hard to see with certitude in the images what the authors claim in the text.

Responses: Thank you for your valuable suggestions. In the revised manuscript, we quantified the numbers of the ACKR1+ vessels and compared it between HT and non-HT patients (Fig. 6F, G). For immunostaining images of CCL21+ fibroblasts and myofibroblasts, there was barely no positive staining in non-HT controls (Fig. 6C), thus we did not provide quantification data. Indeed, for most images, we provided a magnified image in side of it and essential arrows for images and more detailed explanations for figure legends were added to make it easier for readers to catch the points claimed in the text. We hope it is much improved and thank you for your excellent advices.

In the sentence: "Interestingly, we found most of the CCL21-positive cells were distributed in T cell zones of ELS in thyroid tissues of HT patients (Fig. 4D)," The CCL21+ cells are clearly among immune cells, but lack of CD20 expression doesn't automatically mean T cell zone. Can the authors please explain how they arrive to that conclusion?

Responses: The concern of the reviewer about that “lack of CD20 expression doesn't automatically mean T cell zone” is correct. In the revised manuscript, we replaced figure 4D with a new image showing the staining of CD3E (a marker for T cells) to mark the T cell zone of the ELS (in revised Fig. 5D). The image shows a typical structure of germinal centers.

-In the discussion, "Notably, we also found that the number of HEVs were dramatically increased in the thyroid tissues of HT patients compared to non-HT controls" To my knowledge, the authors didn't actually show this.

Responses: Thank the reviewer for his criticism. Indeed, in our previous manuscript, we found the expression of marker genes of ACKR1+ cells (a subset of endothelial cells dominantly distributed in the HEVs) were higher in thyroid tissues from HT patients than that from non-HT patients, which indicated that ACKR1+ vessels were increased in the thyroid tissues from HT patients when compared with no-HT patients. In the revised manuscript, we provided the immunostaining results of HEVs for non-HT controls in figure 6D and 6E. We found that ACKR1+ vessels are much less in non-HT controls than that in HT patients and HEVs are not observed in non-HT controls (revised Fig. 6D-G). As previously reported by others, HEVs are specialized blood vessels first found in lymphoid tissues such as lymph nodes. HEVs are also found in inflammatory tissues or tumors. However, in normal nonlymphoid tissues, it is not observed (Duijvestijn et al. *Am J Pathol.* 1988 Jan;130(1):147-55). (Kabel et al. *J Clin Endocrinol Metab.* 1989 Apr;68(4):744-51)

Minor comments:

- In the sentence: "The infiltrated immune cells accounted for 65% to 82% of all the cells in thyroid tissues of HT patients, most of which were T cells, B cells, and plasma cells." Could these percentages be so high because of technical issues, in which stromal cells die more during the processing of the sample?

Responses: Thank you for your good question. In previous reports, the thyroid tissues of HT patients was characterized by a large scale of lymphocytic infiltration with germinal center formation and destruction of thyroid follicle cells by widespread

apoptosis, accompanied by a variable degree of fibrosis , which is in accordance with the HE staining results in our HT patients (Supplementary Fig.1A), it is not hard to notice from the image that the density of immune cells is much higher than thyroid follicular cells (Supplementary Fig.1A). Thus, the higher percentages of infiltrated immune cells in thyroid tissues of HT patients might be reasonable. It is surprised for us that the percentages of thyrocytes is too low, which might be caused by using dead cell removal kit in single cell isolation, which may remove many thyroid follicular cells that tend to apoptosis.

- "Interestingly, in addition to immune cell subgroups, the expression scores of the top markers for stromal cell subgroups EC2 and F1 identified by Monocle in the thyroid tissues of HT patients were significantly higher than those of non-HT controls, suggesting HEV EC2 endothelial cells and F1 inflammatory fibroblasts are increased in thyroid tissues of HT patients (Fig. S4B)." Can a P-value be provided for this? Visually, the difference doesn't look significant in Fig. S4B.

Responses: Sorry for not describing these details clearly. *P*-value for Fig. S4B was provided in supplementary Tables. In the revised version, it is displayed in supplementary Table 5. The adjusted *P* values are all less than 0.03.

- "Consistently, further investigation of specific markers for EC2 and F1 showed that the expression levels of ACKR1, CCL19, CCL21, CXCL9, SELL, SELP, IL33, and MADCAM1 were significantly increased in thyroid tissues of HT patients (fig. S4C, table S4)." Similarly, what is the P-value for this differential expression of each of these genes? What is the fold change? CCL21, CXCL9 don't seem significantly differentially expressed.

Responses: Thank you for your questions. In the revised manuscript, the P-value and fold change for this differential expression of each of these genes was provided in supplementary Table 5. The expression of *CCL21* and *CXCL9* in HT patients are significantly higher than that in non-HT patients (*q* value= 0.00006 and 8×10^{-8} respectively).

- How does CD38 expression look in the subset that the authors annotated as PCs

compared to other subsets? The expression levels of the marker the authors chose for PCs is not that high compared to other subsets.

Responses: Thank you for your good suggestion. Indeed, both of CD38 (bright) and CD138 are the markers for plasma cells. However, CD138 (SDC1) is a more specific but less sensitive marker. In the revised manuscript, we also added CD38 as a marker for plasma cells in Fig. 1C and Fig. 7G according to the suggestion of the reviewer.

- In the migration assays, what exactly are the error bars showing? Can the authors show the individual datapoints?

Responses: Thank you for your suggestion. In the revised manuscript, we have replaced the histogram of migration assays with a graph showing the individual datapoints. (Revised Fig. 4F, G). The error bars indicate SEM. We have added this explanation to the figure legend.

Reviewer #2 (Remarks to the Author):

The article "Thyroidal stromal and immune cell subsets promote lymphocytic infiltration and thyrocyte destruction in Hashimoto's thyroiditis" describes the results from the first scRNAseq in autoimmune thyroid disease. Results from scRNAseq and the accompanying bioinformatic analysis are complemented by some total RNAseq and immunohistopathology experiments. The work is well done and deserves publication.

Overall it is a bit disappointing that after generating so much data the amount of new information appears modest; the expression of a chemokine receptor (ACKR) that was missed in studies that go back 15 years and a finer dissection of fibroblasts, myofibroblasts and endothelial cells in the high endothelial venule.

The thyroids are clearly very heavily infiltrating and thyroid s cells loss due to sample preparation because the proportion seems too low.

The authors focus the analysis of these stromal cell because they seem to be crucial, as

in canonical secondary lymphoid organs, for then generation of tertiary lymphoid tissue, and some of their result support their proposal, but do not demonstrate it.

Weak points

The lack of control normal thyroid tissue; it is fundamental to support the proposal that the stromal and capillary endothelial cells have acquired special function to generate / maintain autoimmunity, to demonstrate that they are different from the stroma of normal thyroid. Obtaining normal thyroid from nodule resection should be possible.

Responses: Thank for the valuable advice from the reviewer 2. In the revised manuscript, we have added the immunostaining data and found that these special cell subsets are few or absent in thyroid tissues of non-HT controls (Revised Fig. 6). Our results demonstrated that the numbers of ACKR1+ vessels in thyroid tissues are much less in non-HT individuals than that in HT patients while HEVs and CCL21+ fibroblasts or myofibroblasts are not observed in the thyroid tissues of the non-HT individuals.

immunohistochemistry and immunofluorescence images are not top quality. Details will show better if in a frame and magnified inside the picture. The dynamics of cells can nor be directly be inferred from location (line 382), it should say suggest...

Responses: That is a helpful suggestion. We provided magnified images for applicable pictures to make details clearer. In the revised manuscript, we also changed the referred sentence to “CD45+ leukocytes were suggested be undergoing trans-endothelial migration in ACKR1+ venules”

The difficulty in discerning the Th subtypes is not all surprising but Tfh and Tfr should perhaps be looked specifically for.

Responses: Thank you for the valuable comments from the reviewer 2. According to previous literature, follicular regulatory T cells (Tfr) are co-express FoxP3 and CXCR5 while follicular helper T cells (Tfh) express CXCR5, BCL6, PDCD1(PD-1), ICOS or

IL21. We tried to look for Tfh and Tfr by plotting the expression levels of these markers in T cells (Revised supplementary Fig. 5C). It is a pity that cells expressing these marker genes showed no aggregates. 10x single cell RNA-seq combined with membrane protein detection may help to discern Th cell subtypes in future.

Minor

Most people refer to ectopic secondary lymphoid tissue as tertiary lymphoid organs (TLO);

it may be a good idea they use this name

Responses: It is a good idea. We replaced ectopic secondary lymphoid tissue by tertiary lymphoid organs (TLOs) in the revised version of manuscript.

The paper language is in general clear, but abstract and discussion need a bit of editing

Responses: Thank you for your suggestion. We carefully proof-read the manuscript again to improve the writing and to minimize typographical and grammatical errors.

Non standard abbreviations such as infla.mac and infla.DC better avoided.

Line 206, when the authors say “lymphocytic endothelial cells”, the probably mean lymphatic vessels endothelial cells”

Responses: Thank you for your careful correction. We replaced “mac”, “infla.mac” and “infla.DC” by “mø”, “inflam. mø” and “inflam.DC” respectively. “Lymphocytic endothelial cells” has also been changed to “lymphatic vessels endothelial cells”

The use of T8.1 cells should be explained as it is not so common

Responses: Thank you for your reminding. T8.1 is a murine hybridoma T cell line. Explanation and references for the use of T8.1 cells has been provided in the method of revised manuscript.

Legend fig 1, there is no description for abbreviation PB, NaB, GCB (Germinal Center B cells?)

Responses: Thank you for your nice reminding. We have replaced the wrong typed cluster name “PB” by “PC” in figure 1D, and added the description for these abbreviations in the legend of Fig. 1 (NaB: naïve B cells, GCB: Germinal Center B cells).

Reviewer #3 (Remarks to the Author):

The manuscript's major claims are: 1) it is the first report that by using ScRNA-seq identified in the thyroid gland from patients suffering Hashimoto Thyroiditis (HT) the stromal cells that could promote lymphocytic infiltration. 2) identified the cells that will promote the formation of ectopic lymphoid tissue in the thyroid 3) The presence of inflammatory macrophages and dendritic cells in the thyroid. The authors contribute organizing and clustering groups of cells with markers and in this way they identified which cells could be responsible for stimulating the lymphocyte infiltration. Even though this is a tremendous and important contribution to identify the cells that bear the markers for inflammation or for recruiting immune cells, to my knowledge the authors have not been demonstrated that these cells are totally responsible fro the pathogenesis of TH. The authors should be more cautious regarding their conclusion that they proved that these cells will facilitate lymphocyte traffic because even though they bear the markers like CCL21 the in vitro strategy is not enough to prove this. The correct approach will be that the authors should eliminate these cells and prove that there is not lymphocyte infiltration. Therefore I recommend changing the word "prove" in the abstract.

Responses: Thank you for your comments, the advices are correct. In the revised manuscript, we have replaced “prove” by “suggest” in the abstract.

The tables are missing in the manuscript and there are several phrases in the introduction that not have references and they should have them.

Responses: The tables are all uploaded as supplementary files, we will check carefully to make sure all the data are uploaded completely after sending back the revised files. Besides, necessary references have been added in the revised manuscript and we deleted several less important references to meet the requirement of no more than 70 references.

The authors should present results of scRNA-seq regarding which type of antibodies are expression B cells in the thyroid and PBMC. This aspect it is important given that HT is characterized by antibodies against TPO or TG mainly and the question should be where are these cells, in the plasma or in the thyroid.

Responses: It is a meaningful question that we would like to figure out. We could explore the isotype of IgG by plotting the expression of IGHG1/2/3/4 in all the plasma cells and plasmablasts (cell group B6-B8). As showed in following graph, the expression of all isotypes of IgG are higher in thyroid plasma cells and plasmablasts than that in PBMC plasma cells and plasmablasts, which indicated that the antibodies producing ability of the plasma cells and plasmablasts in thyroid tissues of HT patients were higher than that of the PBMCs. The results have been added in the revised manuscript.

However, the scRNA-seq data used in this study was generated by using Single cell 3'library and Gel Bead Kit, which lacks the the 5' mRNA sequence information of BCR genes. Thus, we could not infer which plasma cell could produce the antibodies against TPO or TG.

The results of this manuscript are novel because it is the first work that uses this potent experimental strategy that is scRNA-seq to identify the pattern of molecules that are expressing cells located in the thyroid gland of HT patients. I attached the manuscript where I added my comments. Therefore, I recommend it for publication. This knowledge is relevant because it can be used for the diagnosis and treatment of HT. Moreover, it will also contribute to basic science to help in the characterization of cell types that invaded the thyroid gland and organized ectopic lymphoid structures based on the transcriptome. Yes, the work in this manuscript is convinced and it will be of interest to immunologists and endocrinologists. I attached the manuscript with my comments to improve it.

Thank you for your nice comments.

REVIEWER COMMENTS

Reviewer #1 (Remarks to the Author):

The authors have made substantial improvements to the manuscript; however, I still spot important methodological issues.

The volcano plots shown in figure 8 have extremely inflated P-values given the samples size of 5 individuals. As explained in Zimmerman et al, Nat Comm, 2021 (PMID 33531494), methods such as the one implemented in Monocle have very high type 1 error rates (30-70% for monocle depending on the number of cells). Hence, a generalized linear mixed model controlling for donor should be used for all single cell differential expression analyses.

Furthermore, the authors wrote in the methods that in instances where they applied linear mixed models, they added age and sex as a random effect, these two variables should be added as fixed effects.

In the authors' response to reviewer comments, the authors show they applied Harmony on the immune cells of thyroid and PBMCs using for batch correction (a) donor, or (b) donor and tissue of origin. The two UMAPs look almost identical, but inversed. Can the authors show more plots/metrics to ensure they are applying Harmony correctly? For example, UMAP plots before and after harmony correction, barplots of diversity index in clusters before and after harmony (diversity in donors, and diversity in tissue).

Minor: I believe there is a mistake in the methods about the total number of tests performed wfor the CellPhoneDB analysis.

Reviewer #2 (Remarks to the Author):

The paper is now acceptable for publication

Reviewer #3 (Remarks to the Author):

I agree with this last version of the article. However, I am still with certain doubts regarding the contribution of the manuscript to the field. The authors identified cells that could play a role in lymphocyte trafficking and thyroid destruction in thyroiditis of Hashimoto. However, the authors can only suggest these possible roles. I think the study is very complex; I am not familiar with the techniques. I feel sorry I can not give more advice on it.

Reviewer #1 (Remarks to the Author):

The authors have made substantial improvements to the manuscript; however, I still spot important methodological issues.

1. The volcano plots shown in figure 8 have extremely inflated P-values given the samples size of 5 individuals. As explained in Zimmerman et al, Nat Comm, 2021 (PMID 33531494), methods such as the one implemented in Monocle have very high type 1 error rates (30-70% for monocle depending on the number of cells). Hence, a generalized linear mixed model controlling for donor should be used for all single cell differential expression analyses.

Response: It's very thankful for Reviewer #1's suggestion, we couldn't agree any more. As Zimmerman et al.¹ proposed, an important characteristic of single-cell experiments is that they use many cells from the same individual, and therefore the same genetic and environmental background, could result in the existence of pseudoreplication. So single-cell data have a hierarchical structure, but many current single-cell methods do not address, leading to biased inference, highly inflated type 1 error rates, and reduced robustness and reproducibility.

Hence, Zimmerman et al.¹ proposed applying the generalized linear mixed models (GLMM) with a random effect for individual, to properly account for both zero inflation and the correlation structure among measures from cells within an individual for single cell differential expression analyses.

Zimmerman et al.¹ pointed out in the article that Type I error rates are well-controlled with mixed models, while type I error rates increase with other methods as additional independent samples or more cells are added.

Overall, mixed-effects models lead to the most accurate results when analyzing data with a hierarchical structure²⁻⁴. Therefore, in the last revised version, we have already used the GLMM to control high type I error rates due to comparing cells across very few individuals in Figure 8. We are very sorry that we did not clearly describe that we have applied GLMM to perform differential expression analyses in the Methods section in the first revised version. In this revised draft, we have made detailed supplements to the analysis method in the Methods, and cited the literature referred¹. As we showed below, the *P* values of the almost all genes were dramatically increased after corrected

by GLMM. Moreover, many genes with significant difference between the two subgroups of cells have been filtered after applying GLMM, indicating the GLMM can indeed control the false positive rate to a large extent.

In addition, in this revised version, GLMM has also been applied to all other single cell differential expression analyses (differential expression analysis of single cells of thyroid tissues), and related figures and tables have been revised (Fig1e, Fig S2, Table S2, Table S3).

Inflam. cDC2 vs cDC2

Inflam. mØ vs mØ

1. Zimmerman KD, Espeland MA, Langefeld CD. A practical solution to pseudoreplication bias in single-cell studies. *Nat Commun.* 12(1), 738 (2021).
2. Millar, R. B. & Anderson, M. J. Remedies for pseudoreplication. *Fish. Res.* 70, 397–407 (2004).
3. G. W. Snedecor & W. G. Cochran. *Statistical Methods* (Oxford & IBH Publishing Co., 1994).
4. Tirrell, T. F., Rademaker, A. W. & Lieber, R. L. Analysis of hierarchical biomechanical data structures using mixed-effects models. *J. Biomech.* 69, 34–39 (2018).

2. Furthermore, the authors wrote in the methods that in instances where they applied linear mixed models, they added age and sex as a random effect, these two variables should be added as fixed effects.

Response: Thank you for your reminding. In this revised version, age and sex has been added as fixed effects. The figures and tables have been also revised (Fig S4a, Table S5).

3. In the authors' response to reviewer comments, the authors show they applied Harmony on the immune cells of thyroid and PBMCs using for batch correction (a) donor, or (b) donor and tissue of origin. The two UMAPs look almost identical, but inversed. Can the authors show more plots/metrics to ensure they are applying Harmony correctly? For example, UMAP plots before and after harmony correction, barplots of diversity index in clusters before and after harmony (diversity in donors, and diversity in tissue).

Response: Thank you for your comment. The UMAP plots and diversity index in clusters before and after harmony were shown (Response Fig1 and Fig3). Before harmony correction, the distribution of PBMC cells was also separated with thyroid cells (Response Fig1A). After harmony correction with donor as covariate, the PBMC and thyroid cells were distributed much more evenly (Response Fig1B). The harmony dimensions generated from harmony running using donor as covariate (Response Fig2A) was almost identical with running using donor and tissue as covariates (Response Fig2B) ---tiny difference can be observed when look carefully(pointed by red arrows or circles). But the tiny difference of harmony dimensions causes inversed

UMAP plots (Response Fig1B and C) even setting the same random seed (seed=42), since UMAP algorithm is stochastic. We also provided a 180° rotated graph of Response Fig 1C below to make it easier to compare the difference between Response Fig1B and C. However, the UMAP distances between cells are almost identical.

C (Rotate 180 degrees)

Response Figure.1

(A) UMAP of the merged immune cells before removing batch effect colored by patient ID (left), tissue origin (middle), or cluster (right). (B) UMAP of the merged immune cells after removing batch effect (donor as covariate) colored by patient ID (left), tissue origin (middle), or cluster (right). (C) UMAP of the merged immune cells after removing batch effect (donor and tissue as covariates) colored by patient ID (left), tissue origin (middle) or cluster (right).

Response Figure.2

(A) Harmony dimensions generated from harmony running with donor as covariate. (B) Harmony dimensions generated from harmony running with donor and tissue as covariates.

Response Figure3

(A) Tissue diversity index in clusters before and after harmony. (B) Donor diversity index in clusters before and after harmony

4. Minor: I believe there is a mistake in the methods about the total number of tests performed for the CellPhoneDB analysis.

Response: The interactions in CellPhoneDB are not symmetric. When testing a ligand/receptor pair A_B between clusters X_Y, the expression of partner A is considered within the first cluster (X), and the expression of partner B within the second cluster (Y). Therefore, X_Y and Y_X represent different comparisons and will have different p-values and means.

In our dataset, 18 cell types will generate 324 pairwise comparisons between two cell types including self-comparisons. In CellPhoneDB database, 1189 curated ligand-receptor interactions have been included. A null distribution for each ligand-receptor pair in each pairwise comparison between two cell types will generate a p-value. Thus 385,236 p-values were finally obtained (Supplementary table 4).

Indeed, the current method of analyzing the interaction between cells and cells in single-cell data may be limited, as users, we hope a better and more credible method

for analyzing cell-cell interactions of single cell data could be invented soon.

Reviewer #2 (Remarks to the Author):

The paper is now acceptable for publication

Response: Thank you for your nice comments.

Reviewer #3 (Remarks to the Author):

I agree with this last version of the article. However, I am still with certain doubts regarding the contribution of the manuscript to the field. The authors identified cells that could play a role in lymphocyte trafficking and thyroid destruction in thyroiditis of Hashimoto. However, the authors can only suggest these possible roles. I think the study is very complex; I am not familiar with the techniques. I feel sorry I can not give more advice on it.

Response: Thank you for your kindly comments.

REVIEWER COMMENTS

Reviewer #1 (Remarks to the Author):

I thank the reviewers for making a thorough revision of their computational methods. I have no further comments.